

# Gridded 5-arcmin, simultaneously farm-size- and crop-specific harvested area for 56 countries

Han Su[1,2], Bárbara Willaarts [2], Diana Luna-Gonzalez [2], Maarten S. Krol [1], Rick J. Hogeboom [1,3]

[1] Multidisciplinary Water Management group, Faculty of Engineering Technology, University of Twente, Enschede, 7500AE, the Netherlands
[2] Water Security group, International Institute for Applied Systems Analysis (IIASA), Laxenburg, 2361, Austria
[3] Water Footprint Network, Enschede, 7522NB, the Netherlands

*Correspondence to*: Han Su (h.su@utwente.nl)

**Abstract.**

Farms are not homogeneous. Smaller farms generally have different planted crops, yields, agricultural inputs, and irrigation applications compared to larger farms. However, gridded farm-size-specific data—that is moreover crop specific—is currently lacking. This obscures our understanding of differences between small- and large-scale farms, e.g. with respect to climate change adaptation and mitigation strategies, contribution to (local) food security, and water consumption patterns. This study fills a significant part of the current data gap, by developing high-resolution gridded, simultaneously farm-size- and crop-

specific datasets of harvested area for 56 countries (i.e., covering about half the global cropland). Hereto, we downscaled the most complete global direct measurements of farm size and crop type by compiling state-of-the-art datasets, including crop maps, cropland extent maps, and dominant field size distribution, representative of the year 2010. Using two different crop map sources, we were able to produce two new 5-arcmin gridded datasets on simultaneously derived farm-size- and crop-specific harvested area: one for 11 farm sizes, 27 crops, and 2 farming systems, and one for 11 farm sizes, 42 crops, and 4

farming systems. In line with previous findings, our resulting datasets show major differences in planted crops and irrigated area (%) between farm sizes. Consistency between our resulting datasets and i) observations from satellite images on farm-size-specific oil palm, ii) household surveys on the farm-size-specific irrigated area (%), and iii) previous studies that mapped non-crop-specific farm sizes, support the validity of our datasets. Although at grid level some uncertainties remain to be overcome, particularly those stemming from uncertainties in crop maps, results at country level seem robust. Source data, code,

and resulting datasets are open-access and freely available at https://doi.org/10.5281/zenodo.6976249 (Su et al., 2022).

# 1 Introduction

There are over 608 million farms around the world, which highly vary in their characteristics (Lowder et al., 2016; Lowder et al., 2021). For example, more than 80% of the farms are smaller than 2 hectares and utilize only around 20% of global farmland area of 2.5 billion hectares (Bosc et al., 2013; Lowder et al., 2021). In contrast, the largest 1% of the farms occupy 70% of global farmland area (Lowder et al., 2021). Smaller farms typically apply less irrigation in low- and middle-income countries, making them more vulnerable to water scarcity than larger farms (Ricciardi et al., 2020). In terms of crops and mindful of national differences, smaller farms tend to plant more fruits, pulses, and roots and tubers, while larger farms plant more vegetables, nuts, and oil crops (Herrero et al., 2017; Ricciardi et al., 2018a, b). Furthermore, farmers who operate smaller farms tend to increase the use of non-fixed inputs to increase their productivity, such as fertilizers and pesticides, whereas larger farms rather increase fixed inputs such as machinery (Ren et al., 2019). Whether smaller farms also generate higher yields has long been debated, although it appears that yields often correlate positively with farm size (see Rudra (1968); Savastano and Scandizzo (2017); Gollin (2019); Ricciardi et al. (2021)). What seems undisputed, however, is that smaller farms on average display greater biodiversity than their larger counterparts (Ricciardi et al., 2021; Noack et al., 2021).

Since characteristics vary widely between farms, many studies set out to map the differences, particularly along the dimension of their size to discern small- and large-scale farms (Riesgo et al., 2016; Meyfroidt, 2017). At the global level, farm size mapping was pioneered by Lowder et al. (2016), Samberg et al. (2016), and Fritz et al. (2015). Lowder et al. (2016) estimated the country-level distribution of farm size, based on multiple agricultural censuses. Samberg et al. (2016) used the Mean Agricultural Area (MAA) to assign each sub-national administrative unit with a farm size. A limitation of this approach is that it may overestimate the area of small farms, since being located in an administrative unit dominated by small farms does not necessarily mean that all farms within that unit are indeed small (Ricciardi et al., 2018a, b). Fritz et al. (2015) mapped a gridded global dominant field size distribution, using manually labeled field size data on satellite images and spatial interpolation. The dominant field size distribution by Fritz et al. (2015) was updated by Lesiv et al. (2019). A consequence of interpreting fields as farms, however, is that small farm areas may be overestimated, since large farms can include small-sized fields as well.

Further developments ensued through Herrero et al. (2017), who used the country-level farm size data from Lowder et al. (2016) and Fritz et al. (2015) to develop a dominant farm size map. This map, in turn, was updated by Mehrabi et al. (2020) using the field size distribution from Lesiv et al. (2019). However, despite its improvements, the method employed by Mehrabi et al. (2020) still assigns only one (i.e., a dominant) farm size to each grid cell (5 × 5 arcmin), which reduces its usefulness in estimating the number and area distribution of different farm sizes.

Another important shortcoming in previous studies is that current farm size maps are not crop specific. A potential solution to estimating the planted crops for different farm sizes is to overlap the farm size map with crop maps, e.g. Samberg et al. (2016), Herrero et al. (2017), and Mehrabi et al. (2020). Yet still, such overlays may lead to biases in the assigning of crop-specific areas to farm sizes, because of differences between farm size and MAA, field sizes, and dominant farm sizes, and potentially also due to possible structural differences in crop choices between farm sizes (Ricciardi et al., 2018a, b). In order to address

these limitations, farm-size- and crop-specific datasets would need to be developed simultaneously, which is what Ricciardi et al. (2018a, b) attempted. Arguably the most complete empirical global dataset to day, they collated data from agriculture censuses and household surveys that directly measured crop production or areas in combination with farm size. Their dataset covers about half of the global cropland, including data for 56 countries[1], with subnational data for 46 countries. Still, being defined at administrative unit level, the dataset by Ricciardi et al. (2018a, b) lacks a high-resolution grid-level representation of the data. This resolution gap limits the capability to fulfill the needs of e.g. climate, agricultural and hydrological models which commonly need gridded data as input, which, in turn, obscures our understanding of differences between small- and large-scale farms, e.g. with respect to climate change adaptation and mitigation strategies, contribution to local food security, and water consumption patterns.

This study fills a significant part of the current data gap, by developing high-resolution gridded, simultaneously farm-size- and crop-specific datasets of harvested area for 56 countries, representative of the year 2010. The datasets, moreover, provide additional information on farming systems. To obtain the datasets, we developed and applied a downscaling procedure, in which we used state-of-the-art datasets on field size and crop type, including crop maps (Yu et al., 2020; FAO and IIASA, 2021; Fischer et al., 2021), cropland extent (Latham et al., 2014; Lu et al., 2020), and dominant field size distribution (Lesiv et al., 2019), to downscale the most complete empirical global farm-size- and crop-specific dataset by Ricciardi et al. (2018a, b) from the administrative unit to a 5 arcmin gridded spatial resolution. Two crop maps were used to explicitly consider uncertainties in crop distributions. We validated our resulting datasets using empirical data and comparisons with previous studies.

## 2 Methods

### 2.1 Overview

The gridded, simultaneously farm-size- and crop-specific dataset of harvested areas can be achieved by downscaling the administrative unit level crop-specific farm size structure using gridded crop distribution and gridded dominant field size distribution (Fig. 1). Since certain crops are more prevalent on small farms and others on larger farms as indicated by crop-specific farm size structure, the gridded crop distribution primarily indicate where small and large farms are located. Gridded dominant field size distribution further helps specify the location of small and large farms because, by definition, large fields only belong to large farms and small farms can only be located in small fields. We assumed the best estimation of the farm-size- and crop-specific harvested area distribution is the one that maximizes consistencies with the underlying administrative unit farm-size and grid cell level data.

---

1 Their paper states data is available for 55 countries, but the associated dataset actually contains 56 (Czech Republic seems to be added).

The dataset development involved pre-processing of multiple datasets, establishing optimization for downscaling, and constraints relaxation and solving optimization problems (Fig. 1). The pre-processing included two parts: i) reclassifying crops
to accommodate differences in crop classification used in the underlying datasets and harmonizing the dataset by Ricciardi et al. (2018a, b) and ii) converting the dominant field size distribution into a minimum field area per field size and 5-arcmin grid cell (Sect. 2.2). The downscaling was achieved by maximizing consistencies with multiple datasets that provide information on the location of each farm/field size and planted crops. Specifically, we formulated an optimization for each administrative unit (Sect. 2.3) and solved it via constraints relaxations (Sect. 2.4). Priorities in achieving consistency with the various
underlying datasets were considered during these processes (Sect. 2.3 and 2.4). The spatial crop distribution affects both crop location and farm size location during downscaling and is associated with considerable uncertainties. To consider propagation of such uncertainties, we used two different crop maps, i.e. GAEZv4 (FAO and IIASA, 2021; Fischer et al., 2021) and SPAM2010 (Yu et al., 2020). Doing so allowed us to develop two alternative versions of the final downscaled dataset separately.

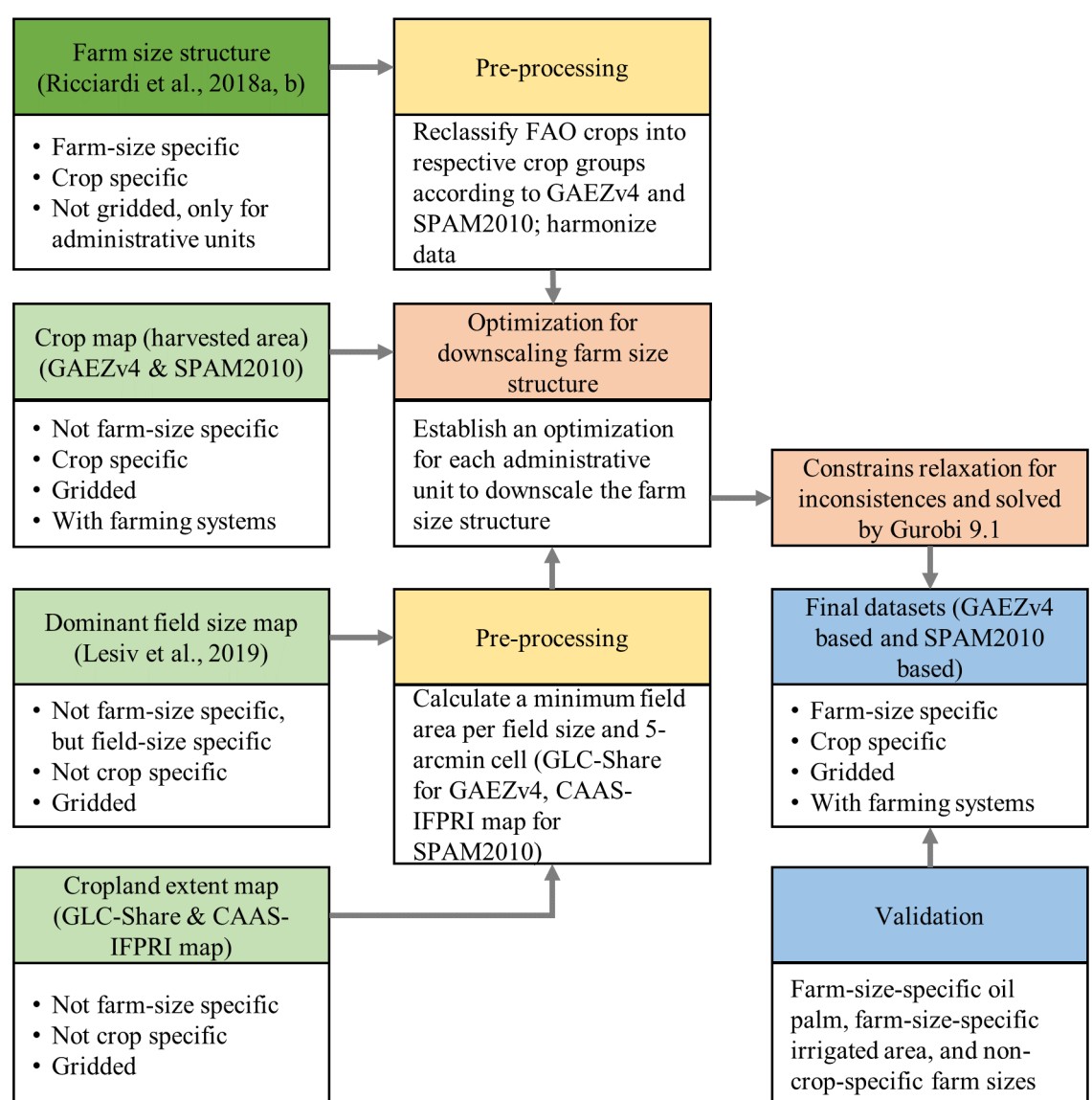

**Figure 1. Diagram of dataset development processors.**

## 2.2 Datasets and pre-processing

The main dataset by Ricciardi et al. (2018a, b) provides the farm-size- and crop-specific cropping area for 56 countries at the administrative unit level (see [S1] for a list of the 56 countries included). The eleven farm sizes in this dataset are based on the classification from the World Census of Agriculture (WCA) (FAO, 2015; Ricciardi et al., 2018a, b; FAO, 2022): 0–1 ha, 1–2 ha, 2–5 ha, 5–10 ha, 10–20 ha, 20–50 ha, 50–100 ha, 100–200 ha, 200–500 ha, 500–1000 ha, and >1000 ha. The cropping area in this dataset indicates either crop area, planted area, harvested area, or cultivated area. Because data quality varies from

country to country and because this dataset was not harmonized in time, we chose to downscale its crop-specific farm size

structure (i.e., the percentage of harvested area per farm size for each crop) instead of the absolute area.

Crop-specific harvested area is taken from two separate crop map sources: GAEZv4 (FAO and IIASA, 2021; Fischer et al., 2021) and SPAM2010 (Yu et al., 2020). These are the most comprehensive crop maps available, containing harvested area of dozens of crops for the year 2010 at 5 arcmin spatial resolution (Kim et al., 2021). GAEZv4 and SPAM2010 have their own crop classification systems, which are given in [S2, S3]. Furthermore, GAEZv4 distinguishes two farming systems, namely

irrigated and rainfed, while SPAM2010 further specifies rainfed into low- and high-input rainfed and subsistence rainfed (in addition to irrigated farming systems).

The dominant field size distribution (Lesiv et al., 2019) indicates where larger farms may be located and contains the spatial distribution for five field sizes: < 0.64 ha, 0.64–2.56 ha, 2.56–16 ha, 16–100 ha, and >100 ha. For pre-processing the dominant field size distribution, cropland extent maps were also included (detailed steps could be found below). All datasets used in this

study are listed in Table 1.

**Table 1. Datasets that were used to develop the gridded, farm-size- and crop-specific datasets of harvested area. \* Here the crop-specific percentage of harvested area per farm size within an administrative unit is meant. \*\* The 27th crop is Fruits and Nuts which is not listed in the document but available in their dataset.**

| Dataset | Indicator | Spatial coverage and resolution | Time | Crop | Note |
|---|---|---|---|---|---|
| Ricciardi et al. (2018a, b) | Farm size structure* | 56 countries; (sub)national administrative unit | Varies from 2001 to 2015 | 154 FAO crops | 11 farm sizes |
| GAEZv4 (FAO and IIASA, 2021; Fischer et al., 2021) | Harvested area (crop map) | Global; gridded, 5 arcmin (10 km) | 2010 | 27 GAEZv4 crops** | 2 farming systems (irrigated and rainfed) |
| SPAM2010 (Yu et al., 2020) | Harvested area (crop map) | Global; gridded, 5 arcmin (10 km) | 2010 | 42 SPAM2010 crops | 4 farming systems (irrigated, low- and high-input rainfed, and subsistence rainfed) |
| Dominant field size distribution (Lesiv et al., 2019) | Dominant field size | Global; gridded, 30 arcsec (1 km) | Varies from 2000 to 2017 | Not crop specific | 5 field sizes |
| GLC-Share (Latham et al., 2014) | Cropland extent | Global; gridded, 30 arcsec (1 km) | Around 2010 | Not crop specific | The based map of GAEZv4 |

| CAAS-IFPRI cropland extent map (Lu et al., 2020) | Cropland extent | Global; gridded, 15 arcsec (0.5 km) | 2010 | Not crop specific | The base map of SPAM2010 |
|---|---|---|---|---|---|

To pre-process the dataset by Ricciardi et al. (2018a, b), we first reclassified their crops (who followed the FAO classification) into 27 GAEZv4 crops and 42 SPAM2010 crops, respectively. Crop reclassification details can be found in [S2, S3]. We used the cropping area to obtain the crop-specific farm size structure. In this dataset, the cropping area is crop specific and includes four items: crop area, planted area, harvested area, and cultivated area. These variables were identified by the dataset by Ricciardi et al. (2018a, b) from the local agriculture census. There is no worldwide standard definition for these items (FAO, 2015), meaning local agriculture censuses can apply their own preferred definitions. In general, however, *planted area* is used for temporary crops; *cultivated area* for temporary crops and permanent crops; *crop area* for temporary crops, permanent crops, fallow fields, meadows, and pastures; and *harvested area* for the cultivated area excluding the area rendered unsuitable for cultivation by natural disasters or other reasons (FAO, 2015, 2020). In terms of data availability, one or two of these items are available for most countries at the least. If more than one item was available, we harmonized the data by taking the item with the largest overall area (after crop reclassification) to estimate farm size structure, since a larger overall area typically means that more farm size classes have available data. If none of the four items was available, we used crop production data provided by the dataset by Ricciardi et al. (2018a, b) as a proxy for the crop-specific farm size structure, assuming constant yields across farm sizes.

During pre-processing we also converted the $1 \times 1$ km dominant field size distribution map into a minimum field area per field size and 5-arcmin grid cell to align with the spatial resolution of crop maps. We interpreted *dominant field size* as that fields of that size accounting for at least 50% of cropland in the grid cell. For each field size, we calculated the minimum field area for each 1-km cell by using the 50% of cropland extent. We then summed the minimum field area from 1-km to 5-arcmin cells and scaled the summed area to cover 50% of croplands in 5-arcmin cells. The minimum field area of field size 16–100 ha is 120 ha in a 5-arcmin cell which means, for example, farms larger than 16 ha should occupy at least 120 ha in the cell. To keep cropland extent consistent with the crop maps during downscaling, GLC-Share was used with the GAEZv4 crop map, while we used CAAS-IFPRI cropland extent map with the SPAM2010 crop map.

**2.3 Optimization for downscaling**

For each administrative unit defined in the dataset by Ricciardi et al. (2018a, b), we established the following optimization problem for our downscaling procedure. Note that the dataset by Ricciardi et al. (2018a, b) identifies eleven farm sizes and the dominant field size distribution (Lesiv et al., 2019) identifies five field sizes.

**Sets:**

$c$, Crops

$f$, Farm size, labelled by the lower bound of the eleven farm sizes

$e$, Field size, labelled by the lower bound of the five field sizes

$s$, Farming system

$a$, Administrative unit

$g$, Grid cell

**Parameters:**

$ha.R_{c,f,a}$, Crop-specific farm size structure, percentage of the harvested area of farm size $f$ that plant crop $c$ in the administrative unit $a$ (from the dataset by Ricciardi et al. (2018a, b))

$ha.S_{c,s,g}$, Harvested area of crop $c$ under farming system $s$ at grid cell $g$ (from crop map, either SPAM2010 or GAEZv4)

$ha.L_{e,g}$, Minimum field area of field size $e$ at grid cell $g$ (from dominant field size distribution by Lesiv et al. (2019) and crop extent map by Latham et al., (2014) and Lu et al., (2020)

$p_f$, The minimum farm area of farm size $f$ in any grid cell when the farm size $f$ exists, i.e., the lower bound of the farm size class $f$

$l$, Elastic factor

**Variables:**

$ha_{c,f,s,g}$ Harvested area of crop $c$, farm size $f$, farming system $s$ in grid cell $g$ (estimated by this study)

**Objective function:**

Since we aim to downscale the dataset by Ricciardi et al. (2018a, b), we maximized—within the constraints— consistencies with the dataset by Ricciardi et al. (2018a, b):

$$min \sum_{c,f} abs\left( ha.R_{c,f,a} \sum_{s,g\in a} ha.S_{c,s,g} - \sum_{s,g\in a} ha_{c,f,s,g} \right) \qquad (1)$$

**Constraints:**

The first constraint is meant to ensure consistency with the respective crop maps we used and states that the total harvested area per crop per farming system per grid cell in our datasets must be equal to the harvested area per crop per farming system per grid cell in the respective crop map.

$$\sum_f ha_{c,f,s,g} = ha.S_{c,s,g}, \forall c, s, g \qquad (2)$$

The second constraint requires a minimum level of consistency with the dataset by Ricciardi et al. (2018a, b) and states that the relative difference in farm size structure between our estimation and the dataset by Ricciardi et al. (2018a, b) cannot be more than 10%. This constraint ensures that, even when other constraints are hard to meet, we do not diverge too far from the dataset by Ricciardi et al. (2018a, b). This constraint takes priority over the following constraints, meaning we would relax other constraints to meet this one. The 10% relative difference mark is an educated guess based on timestamp differences in the dataset by Ricciardi et al. (2018a, b) and overall assumed uncertainties underlying each of the datasets.

$$90\% * ha.R_{c,f,a} \sum_{s,g \in a} ha.S_{c,s,g} \leq \sum_{s,g \in a} ha_{c,f,s,g} \leq 110\% * ha.R_{c,f,a} \sum_{s,g \in a} ha.S_{c,s,g}, \forall c, f \qquad (3)$$

The third constraint sets a minimum allocated area for each farm size in each grid cell if the farm size exists in the cell. This minimum allocated area is not necessarily required by the definition of farm size, yet we reckoned it is still reasonable to include because a 5-arcmin grid cell is mostly much larger than a single farm. Given both the presence of uncertainties in these constraints and inconsistencies among datasets used, we incorporated this constraint in a hard form and soft form during the optimization: we used the hard form by default, but transitioned to the more relaxed soft form when the optimization was infeasible (see also Sect. 2.4). Note that the soft form does not strictly require a minimum allocation area for each farming system.

Hard form:

$$ha_{c,f,s,g} \geq p_f, \forall c, f, s, g, if\ ha_{c,f,s,g} > 0 \qquad (4)$$

Soft form:

$$\sum_s ha_{c,f,s,g} \geq l \times p_f, \forall c, f, g, if\ ha_{c,f,s,g} > 0 \qquad (5)$$

The fourth constraint sets a minimum area for certain farm sizes according to the spatial distribution of dominant field size. The rationale is that a field can only belong to a farm equal or larger than its own size. We assumed a uniform distribution of area within each farm size, like Ricciardi et al. (2018a, b), to accommodate the different classifications of size in farms and fields. For example, 40% of the area in the farm size 10–20 ha was assumed to be in 16–20 ha class in Eq. (7).

For field areas and farms larger than 100 ha:

$$\sum_{c,s,f \geq 100} ha_{c,f,s,g} \geq ha.L_{100,g}, \forall g \qquad (6)$$

For field areas larger than 16 ha and farms larger than 10 ha:

$$\sum_{c,s,f \geq 20} ha_{c,f,s,g} + \frac{20 - 16}{20 - 10} \sum_{c,s} ha_{c,10,s,g} \geq ha.L_{100,g} + ha.L_{16,g}, \forall g \qquad (7)$$

For field areas larger than 2.56 ha and farms larger than 2 ha:

$$\sum_{c,s,f \geq 5} ha_{c,f,s,g} + \frac{5 - 2.56}{5 - 2} \sum_{c,s} ha_{c,2,s,g} \geq ha.L_{100,g} + ha.L_{16,g} + ha.L_{2.56,g}, \forall g \qquad (8)$$

For field areas larger than 0.64 ha and any farm size:

$$\sum_{c,s,f \geq 1} ha_{c,f,s,g} + \frac{1 - 0.64}{1 - 0} \sum_{c,s} ha_{c,0,s,g} \geq ha.L_{100,g} + ha.L_{16,g} + ha.L_{2.56,g} + ha.L_{0.64,g}, \forall g \qquad (9)$$

Since areas should not assume negative values, we also include non-negative area constraints:

$$ha_{c,f,s,g} \geq 0, \forall c, f, s, g \qquad (10)$$

## 2.4 Constraints relaxation and solving procedures

When the above optimization (Eq. (1)–(10)) proved infeasible, we first replaced the hard form of minimum allocated area (i.e., the third constraint) (Eq. (4)) for all farm sizes with the soft form (Eq. (5)) and applied the elastic factor with the following values in order: 1, 1/2, 1/4, 1/8, 1/16, 1/32, 1/64, and 0. If optimization was still infeasible, we relaxed the minimum area constraint required by the dominant field size distribution (i.e., the fourth constraint) by removing the constraints from large to small farms until the optimization was feasible. Relaxing the minimum area constraint did not happen often during downscaling.

Once the above optimization became feasible, we did not necessarily strike a unique global optimum. Therefore, we calculated up to 80 (sub)optimal solutions with the same level of consistencies and averaged these to obtain the final solution. Since the number and quality of solutions depend on the searching process of the solving toolbox, this procedure may still leave some bias in the final averaged solution.

Each optimization was solved by Gurobi v9.1, a fast commercial optimization solver, using the dual simplex method (Gurobi Optimization, 2021). Optimization was taken as infeasible by the solver's initial evaluation or if it is computationally unsolvable (cannot be solved within 150 seconds). Most of the optimal solutions were obtained within 60 seconds when feasible. For those administrative units that contained more than 300 5-arcmin grid cells, the optimization becomes highly complex. This posed a challenge for the solver, with the number of decision variables increasing to over half a million. As a workaround, we applied a two-tiered optimization, where we first divided all grid cells randomly into several groups. Each group included ~100 grid cells (except for Russia, where groups were set to contain ~200 grid cells to keep the total number of groups below 300). Next, we solved the optimization at group level, followed by solving it at the cell level within each group. Out of 3,421 administrative units, 244 units underwent this workaround procedure, collectively covering 89.4% of grid cells in this study. The entire optimization was performed on a desktop computer (Intel(R) Core(TM) i7-8700 CPU @ 3.20GHz, RAM 16 GB) taking 9 days.

Finally, we masked the crop-specific farm size as unknown if their crops are not covered by the dataset by Ricciardi et al. (2018a, b). For these crops, the optimization would still estimate their farm size structures only based on the distribution of crops and dominant field size. Since the overall farm size structure is absent and dominant field size is not sufficient to estimate all farm sizes, uncertainties of these crops are significantly larger than those associated with the dataset by Ricciardi et al. (2018a, b).

## 2.5 Validation and comparison with previous studies

Ideally, we would have validated our farm-size- and crop-specific datasets with observations. However, there are limited empirical datasets available, and if there are, most are not farm-size specific. Given these limitations, we validated our datasets using two empirical datasets. The first is by Descals et al. (2020), who developed a global gridded farm-size-specific oil palm map using deep learning and satellite images for the year 2019. We validated our datasets for five countries that are covered

by both our datasets and the dataset by Descals et al. (2020) (Fig. A1). To interpret their size classification, we adopted the definition of small oil palm farms by Indonesia (the world's largest palm oil producer and exporter) mentioned by Descals et al. (2020), who apply a 25 ha threshold to distinguish small from large farms, i.e., between the two scales as included in Descals et al. (2020). We calculated the Pearson correlation coefficient at grid cell level (i.e, 5 arcmin) and two additional spatial scales, i.e., 15 arcmin and 25 arcmin, using a spatial moving average. We validated our GAEZv4 and SPAM2010 crop map based datasets, separately.

The second empirical dataset to which we compared our datasets is that of farm-size-specific percentage of irrigated area at the country level from the FAO RuLIS (Rural Livelihoods Information System) database (FAO, 2021). RuLIS includes micro-level household survey data representative of the year 2010. Eleven out of 56 countries included in our study are also listed in RuLIS (see an overview in [S4]). Based on these household surveys, we calculated the percentage of total irrigated area (irrigated area divided by cultivated area) for each farm size (classified by crop area) where at least five survey samples are available. Once more, we calculated the correlations between our estimates and those derived from the household surveys. Although this validation considers farm-size-specific farming systems, the data is aggregated over crops.

To further validate our datasets, we compared our datasets to two other studies. The first is by the FAO and has just been published (FAO, 2022). This dataset contains structural data obtained through agricultural censuses, including total crop areas per farm size, at country level, for the years 1990, 2000, and 2010. We compared our datasets with the structural data of 2010 (the year our datasets are most representative of), and complementary with data of the year 2000 as well. The reason to include data on 2000 too is that data does not rely so heavily on interpolation as does 2010 (FAO, 2022), making the comparison more robust although temporal representativeness is less appropriate. Another advantage of including FAOSTAT structural data of 2000 is that it allows for the comparison with the widely used dataset by Lowder et al. (2016) at the same time since the dataset by Lowder et al. (2016) is largely the same as FAOSTAT structural data of 2000 [S5].

The second study to which we compared our datasets is by Mehrabi et al. (2020), who mapped geographic distributions of farm sizes. The dataset by Mehrabi et al. (2020) uses the same farm size distribution as the dataset by Lowder et al. (2016) at the country level, but adds the dominant farm size at 5-arcmin grid cell level. For our comparison, we calculated—at grid cell level—the dominant farm size from our datasets with the farm size that operates the largest total harvested area per grid cell, for our GAEZv4 and SPAM2010 crop map based datasets, separately.

## 3 Results

### 3.1 Dataset statistics

#### 3.1.1 Crop types and farm sizes

We identified gridded harvested area for 56 countries, 11 farm sizes, 27 crops and 2 farming systems based on the GAEZv4 crop map, and for 42 crops and 4 farming systems based on the SPAM2010 crop map, both at 5-arcmin spatial resolution. Fig. 2 illustrates the harvested area of rainfed maize belonging to two farm sizes (2–5 ha and 500–1000 ha) according to our farm-size- and crop-specific harvested area dataset based on the GAEZv4 crop map. Statistics of crop type and farm size show the prevalence of certain crop groups for certain farm sizes (see [S2] for the crop groupings of the 27 GAEZv4 crops). Fig. 3(a) shows that, as farm size increases, oil crops and fodder crops become more prevalent, while fruits and nuts, pulses, and roots and tubers become less widespread. Our dataset based on the SPAM2010 crop map shows comparable results to that based on GAEZv4 (see Fig. A2 for crop groupings as per [S3]). These statistics are consistent with earlier findings by Ricciardi et al. (2018a, b) and Herrero et al. (2017).

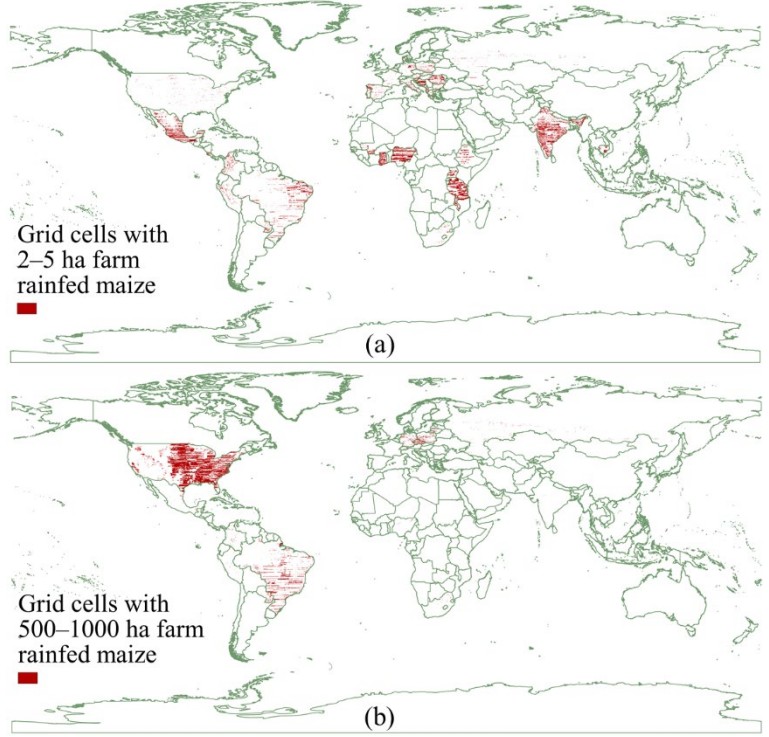

**Figure 2. Grid cells with a harvested area of rainfed maize on farm size 2–5 ha (a) and farm size 500–1000 ha (b), according to our farm-size- and crop-specific dataset based on the GAEZv4 crop map.**

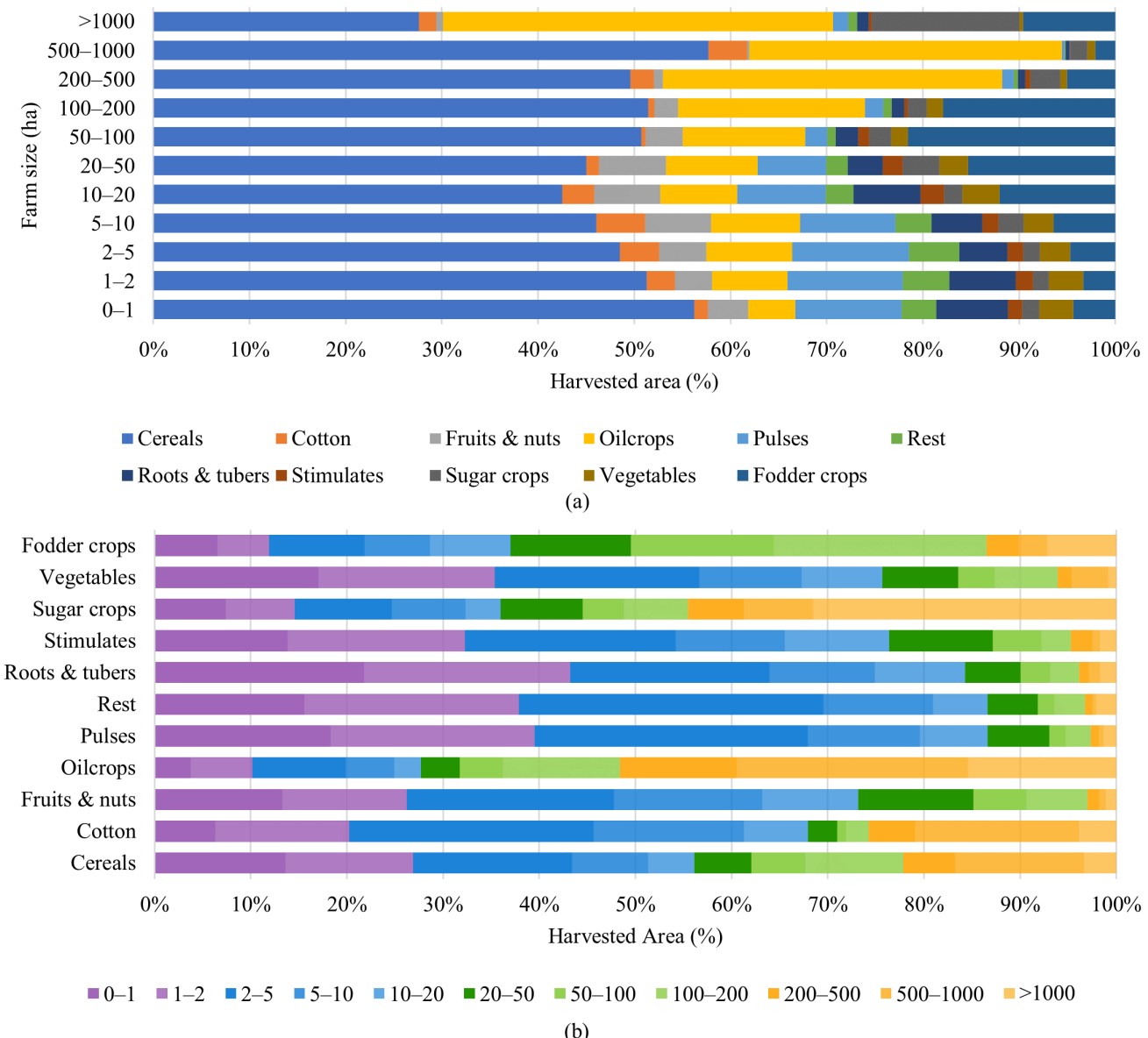

Figure 3. Harvested area of crop groups within each farm size (a) and harvested area of crop groups by farm size (b) according to our farm-size- and crop-specific harvested area dataset based on the GAEZv4 crop map. The alternative version based on SPAM2010 crop map is given in Fig. A2.

### 3.1.2 Farming systems and farm size

Besides providing farm-size- and crop-specific harvested areas, we added information on farming systems inherited from crop maps. Statistics of farming system and farm size derived from our dataset reveal that small farms irrigate a larger relative share of their harvested area than large farms (Fig. 4, Fig. 5), which aligns well with earlier ones by Ricciardi et al. (2020). Here, the

finding is not sensitive to the threshold used to set apart small from large farms, whose possible values can range from 1 ha to 42 ha as suggested by Khalil et al. (2017) and FAO (2017, 2019). Note, however, that this alignment does not hold for some countries (see Sect. 3.2.2 for further details).

Our dataset based on the SPAM2010 crop map further divides rainfed farming systems into low-input, high-input, and subsistence rainfed systems (Fig. 4(b)). Associated statistics show a clear correlation between low-input and subsistence rainfed farming systems and smaller farm sizes. At the same time, smaller farms do not consist exclusively of low-input and subsistence rainfed farming systems, since these smaller farms also operate a sizable portion of the irrigated and high-input rainfed area (see Fig. 4(b)). Similarly, the predominant farming system type of larger farms is high-input rainfed, but high-

input rainfed systems are not solely employed at larger farms.

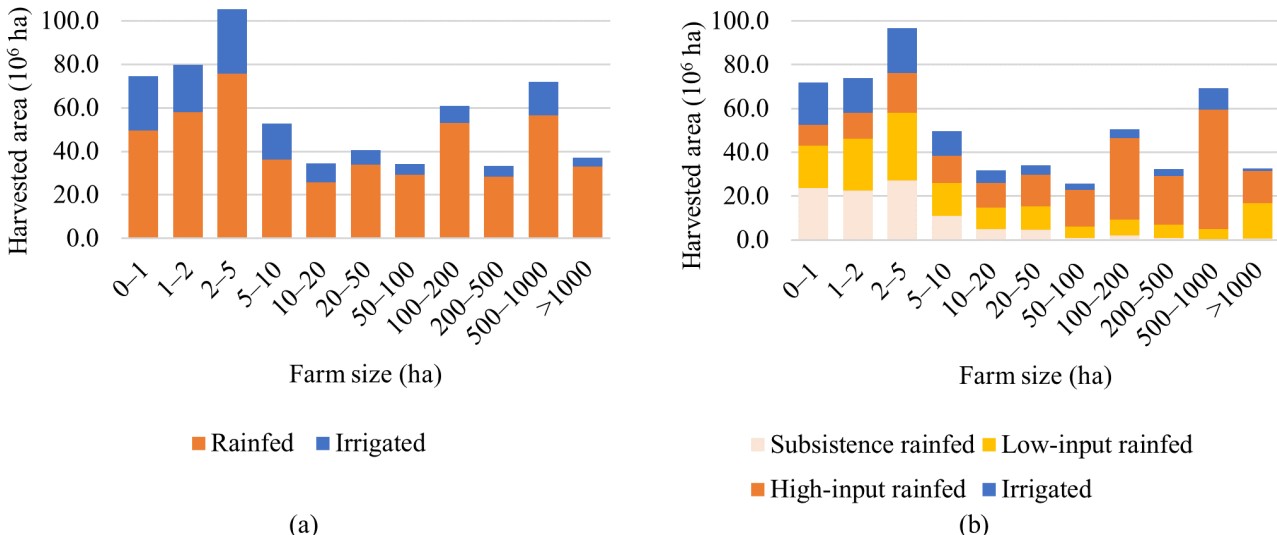

(a)                                                                                  (b)

**Figure 4. The distribution of irrigated and rainfed farming systems per farm size according to our farm-size- and crop-specific harvested area datasets based on the GAEZv4 crop map (a) and the SPAM2010 crop map (b). Note, SPAM2010 further divides rainfed farming system into low-put, high-input, and subsistence rainfed farming systems.**

To further explore irrigation practices, we overlapped our datasets with the annual average blue water scarcity map by Mekonnen and Hoekstra (2016), who classified water scarcity in four categories, i.e., low, moderate, significant, and severe water scarcity. This analysis also confirms an earlier finding by Ricciardi et al. (2020) that even though small farms irrigate a larger relative share of their area than large farms *on average*, large farms irrigate a larger relative share than small farms when water is scarce (Fig. 5). Fig. 5 shows a relatively low irrigation share for farms >1000 ha which would undermine this finding.

However, the total relative irrigation share of large farms is still larger than that of small farms, because this farm size makes up less than 4.5% of large farms located in water scarce areas. Note, that the main aim of Fig. 5 is to compare statistics of our datasets with previous studies instead of drawing conclusions on irrigation levels for specific farm sizes, which may need further investigation on influencing factors and uncertainties.

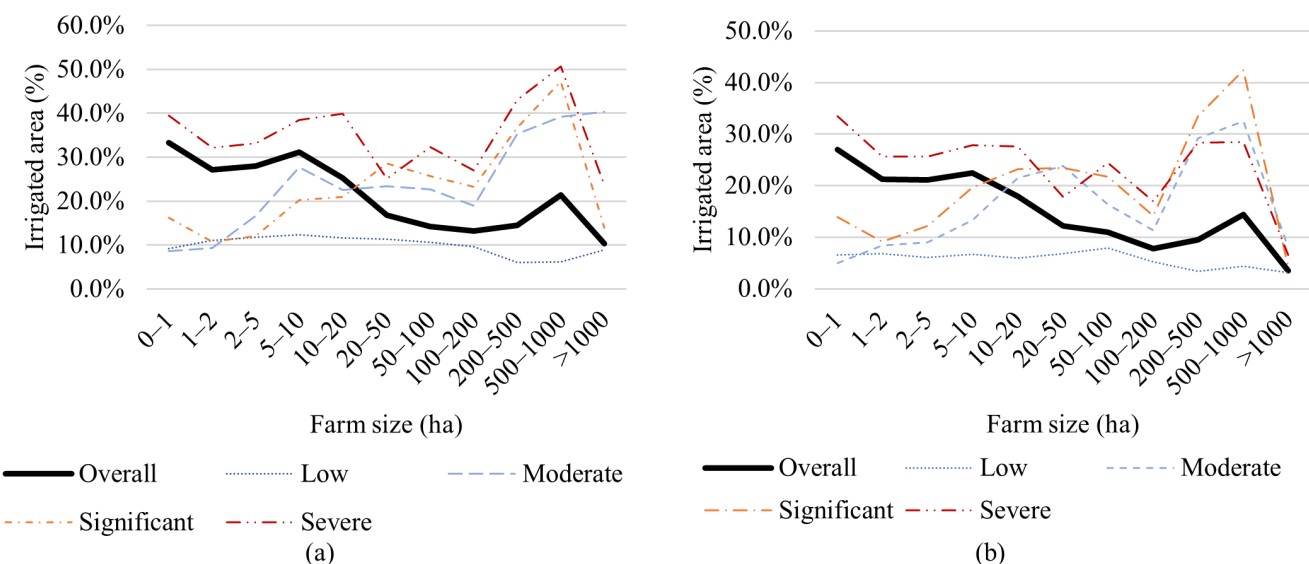

**Figure 5. The percentage of irrigated area by farm size according to our farm-size- and crop-specific harvested area datasets based on the GAEZv4 crop map (a) and the SPAM2010 crop map (b) under four blue water scarcity levels (WSL) by Mekonnen and Hoekstra (2016). Low blue WSL indicates blue water consumption does not exceed blue water availability; moderate WSL indicates blue water consumption is 100–150% of blue water availability; significant WSL indicates blue water consumption is 150–200% of blue water availability; and severe WSL indicates blue water consumption is larger than 200% of blue water availability.**

## 3.2 Validation

### 3.2.1 Validation with empirical data on farm-size-specific oil palm harvested area

Table 2 shows that validation with farm-size-specific oil palm data yields a significant positive correlation in most countries, for both small and large farms. At larger spatial scales, the correlation becomes stronger, indicating that the subnational distributions of oil palm harvested area in our datasets are similar to those of Descals et al. (2020). Besides the threshold of 25 ha to set apart small from large farms, we repeated the comparison with 10 ha and 50 ha thresholds which resulted in similar correlations (see [S6, S7] for detailed results of these comparisons). This indicates that, at least for oil palm, found relations are not sensitive to the choice of threshold.

Despite strong overall correlations, we observed some differences for certain regions, particularly Costa Rica and the United Republic of Tanzania. Some of these differences can be attributed to inconsistencies between harvested area according to the crop maps we used and the validation dataset. We compared total oil palm area according to the crop maps we used and the validation dataset, and found that if the oil palm locations in the crop maps differed from the validation map (no significant positive correlation), farm-size-specific validation was poor as well (Table 2). This implies that the accuracy of our estimates of farm-size- and crop-specific harvested area is limited by the accuracy of oil palm locations in crop maps. The (minor) differences between validation results for the GAEZv4 based dataset and the SPAM2010 based dataset can also largely be attributed to the same reason.

**Table 2. Pearson correlation coefficient between the harvested area of oil palm estimated by satellite images from Descals et al. (2020) and i) GAEZv4 crop map based farm-size- and crop-specific dataset (Gb) and ii) SPAM2010 crop map based farm-size- and crop-specific dataset (Sb), respectively, for small farms (<25 ha), large farms (≥25 ha), and all farms at various spatial resolutions. All farms compared the oil palm area in GAEZv4 and SPAM2010 crop map, whose results imply the accuracy of our estimates of farm-size- and crop-specific harvested area is limited by the accuracy of oil palm locations in crop maps. * p<0.005. ** p<0.001.**

| | | Small farms | | | Large farms | | | All farms | | |
|---|---|---|---|---|---|---|---|---|---|---|
| | | 5 arcmin | 15 arcmin | 25 arcmin | 5 arcmin | 15 arcmin | 25 arcmin | 5 arcmin | 15 arcmin | 25 arcmin |
| Colombia | Gb | 0.177* | 0.313** | 0.397** | 0.112** | 0.238** | 0.334** | 0.232** | 0.374** | 0.465** |
| | Sb | 0.218** | 0.547** | 0.684** | 0.385** | 0.620** | 0.701** | 0.409** | 0.652** | 0.729** |
| Costa Rica | Gb | 0.086 | 0.183** | 0.215** | -0.012 | -0.074 | -0.144** | 0.032 | 0.001 | -0.043 |
| | Sb | 0.836** | 0.944** | 0.971** | 0.771** | 0.891** | 0.925** | 0.877** | 0.925** | 0.929** |
| Brazil | Gb | 0.245** | 0.396** | 0.483** | 0.177** | 0.258** | 0.271** | 0.326** | 0.398** | 0.423** |
| | Sb | 0.133** | 0.190** | 0.248** | 0.087** | 0.091** | 0.084** | 0.148** | 0.154** | 0.156** |
| United republic of Tanzania | Gb | 0.01 | -0.109* | -0.202** | -0.011 | -0.039 | -0.063 | 0.022 | -0.115* | -0.218** |
| | Sb | 0.024 | 0.025 | 0.069 | | | | 0.022 | 0.014 | 0.065 |
| Peru | Gb | 0.172** | 0.350** | 0.438** | 0.024 | 0.139** | 0.237** | 0.111** | 0.263** | 0.363** |
| | Sb | 0.367** | 0.389** | 0.429** | 0.141** | 0.216** | 0.240** | 0.302** | 0.395** | 0.436** |

### 3.2.2 Validation with empirical data on farm-size-specific irrigation estimates

Fig. 6 shows that our datasets are quite consistent with empirical data on farm-size-specific irrigation estimates in terms of country-level farm-size-specific percentage of irrigated area. More detailed results in [S8] further illustrate how our datasets capture the higher percentage of irrigated areas as indicated by the household surveys in both small and large farms in most countries. However, we also found that our datasets systematically underestimate the percentage of the irrigated area with respect to these same household surveys, both in our GAEZv4 and SPAM2010 based datasets of harvested areas. Fig. 6(c) and 6(d) show that these underestimations are still present if we compare the percentage of irrigated area for all farms from the crop maps. This systematic underestimation may therefore be explained by different measurements of irrigated area and cultivated area in the validation dataset compared to the crop maps.

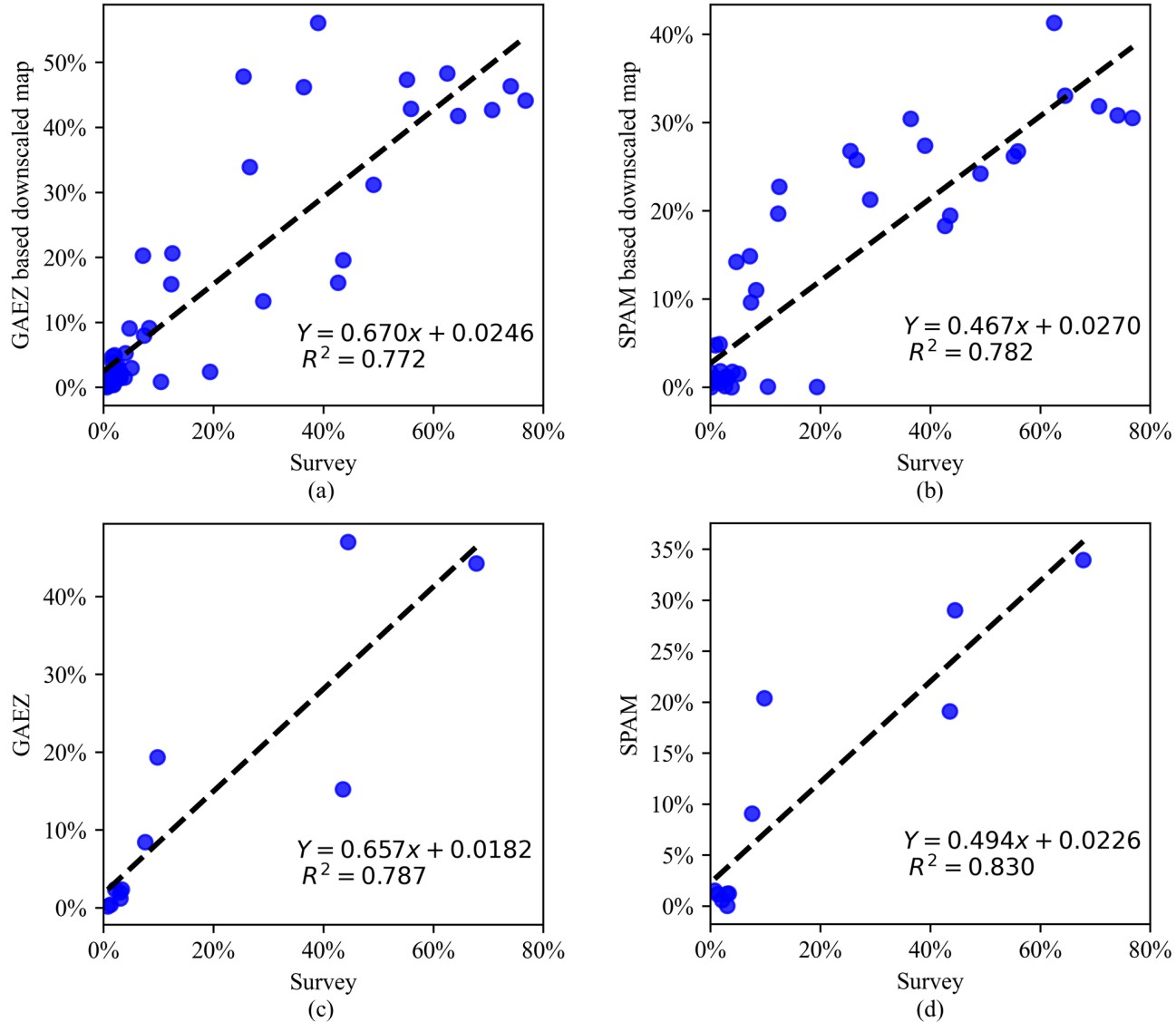

**Figure 6. Correlations on the farm-size-specific irrigated area (% of total harvested area per farm size) between household survey data from FAO RuLIS (Rural Livelihoods Information System) database (FAO, 2021) and our GAEZv4 based (a) and SPAM2010 based, farm-size- and crop-specific datasets of harvested area (b), and correlations on the irrigated area of all farms (% of the total harvested area) between household survey data from FAO RuLIS (Rural Livelihoods Information System) database (FAO, 2021)**
**and GAEZv4 (c) and SPAM2010 (d), all for eleven countries.**

### 3.2.3 Validation through comparison with other studies

Finally, we compared our high-resolution farm-size- and crop-type specific harvested area datasets with FAOSTAT, whose structured data contains farm size structures of 44 overlapping countries for the years 2000 and 2010 (FAO, 2022). Results show that (non-crop-specific) farm size structures of our datasets are similar to FAOSTAT structure data for most countries.

Fig. 7 and Fig. A3 show the large similarities of farm size structures of 28 countries for 2010, while of the remaining 16 countries, farm size structures of Brazil, Czechia, Ethiopia, Germany, Greece, Poland, and Portugal show good correspondence for 2000. The latter also implies these estimates are similar to the dataset by Lowder et al. (2016).

Not all countries' farm size structure corresponds well between the datasets. Farm size structure according to our datasets for Albania, for example, lies in between the FAOSTAT data for 2000 and 2010, and our datasets farm size structures of Costa Rica, Lithuania, and Mexico also deviate slightly from the FAOSTAT structure data. One explanation for such differences could come from how different datasets harmonize collected data into a farm size classification system. For example, if only farm sizes >100 ha are reported, areas could be classified into farm sizes 100–200 ha or be redistributed to farm sizes 100–200 ha, 200–500 ha, and so on. However, the farm size structure of our datasets is inherited from the dataset by Ricciardi et al. (2018a, b), which in turn was based on highly similar local agricultural census and household surveys which FAOSTAT likewise drew from.

While decent overall correspondence between our datasets and either FAOSTAT 2000 or 2010 data might be sufficient grounds to validate our estimates on farm size structure, and particularly correspondence to 2010 being the reference year for our datasets, it should be noted that farm size structures of several countries changed significantly between 2000 and 2010, e.g. Bulgaria and Germany, a period of just 10 years. The FAO themselves also indicate that the robustness of their 2010 estimates is fragile, in part due to significant usage of interpolation (FAO, 2022). Moreover, for 5 of the 44 analyzed countries (i.e., Burkina Faso, Colombia, Peru, and Russian Federation), it remains unclear what causes these differences.

Comparing our datasets with the dataset by Mehrabi et al. (2020), Fig. 8 shows that the patterns of spatial distributions of dominant farm sizes are similar across both datasets. For the farm-size- and crop-specific dataset based on the GAEZv4 crop map, 54% of grid cells' dominant farm sizes correspond to those in the dataset by Mehrabi et al. (2020), while 28% are larger, and 18% are smaller. For the SPAM2010 based counterpart, 53% of grid cells' dominant farm sizes are similar to the dataset by Mehrabi et al. (2020), while 26% are larger, and 21% are smaller. Here, similar means the farm size in our datasets is the same or next to the farm size in the dataset by Mehrabi et al. (2020). [S9] provides a more detailed analysis of this comparison. As shown in Fig. 7, there are still differences between our datasets and the dataset by Lowder et al. (2016) (FAOSTAT structure data of 2000). These differences can also be seen in the comparison with the dataset by Mehrabi et al. (2020) since the dataset by Mehrabi et al. (2020) keeps the same country level farm size distribution as the dataset by Lowder et al. (2016). Note, that the comparison of dominant farm size may magnify the differences in farm size structure between our datasets and the dataset by Mehrabi et al. (2020) since the dominant farm size in the dataset by Mehrabi et al. (2020) may be the second-dominant farm size in our datasets.

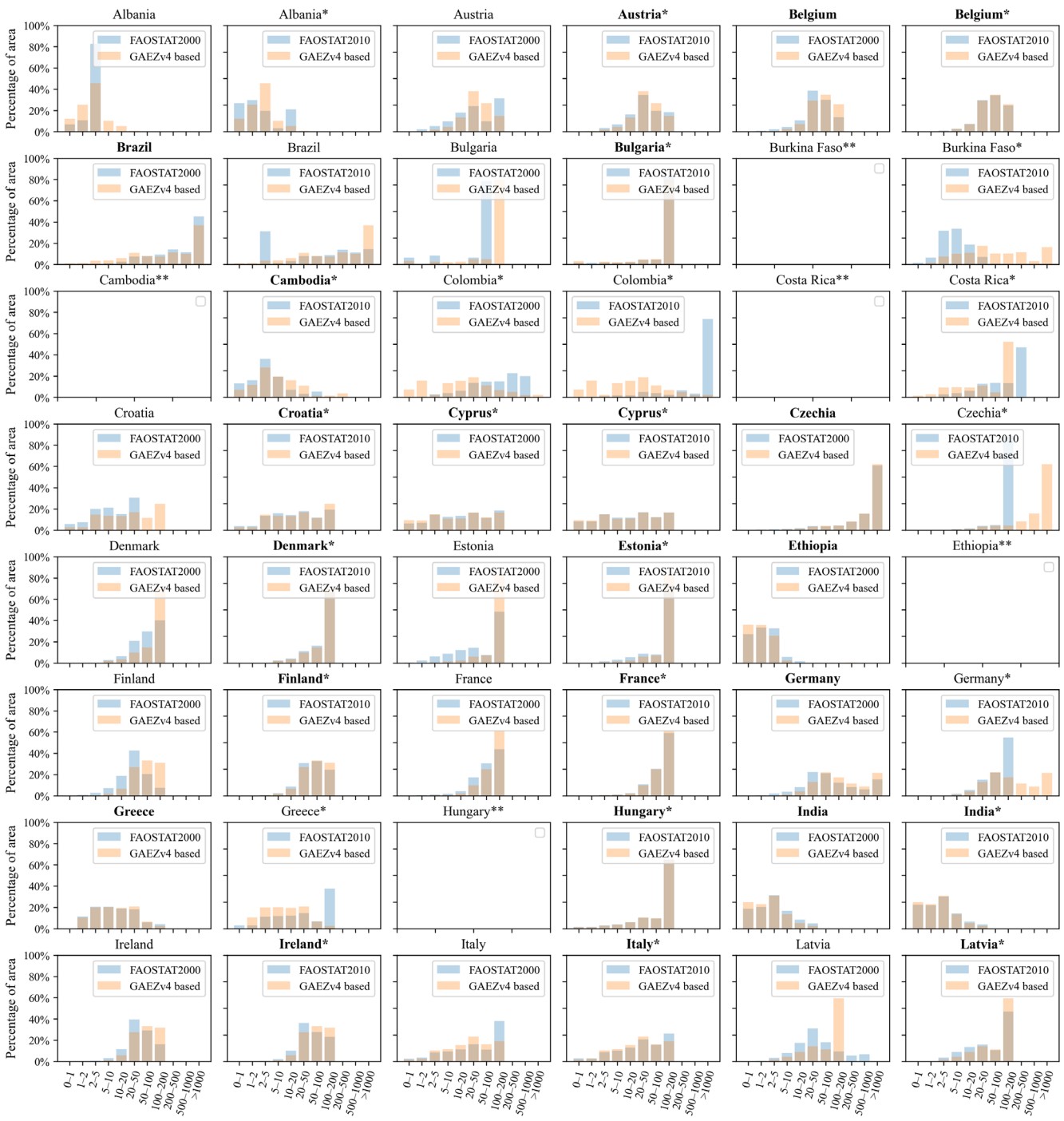

(To be continued)

(Continued)

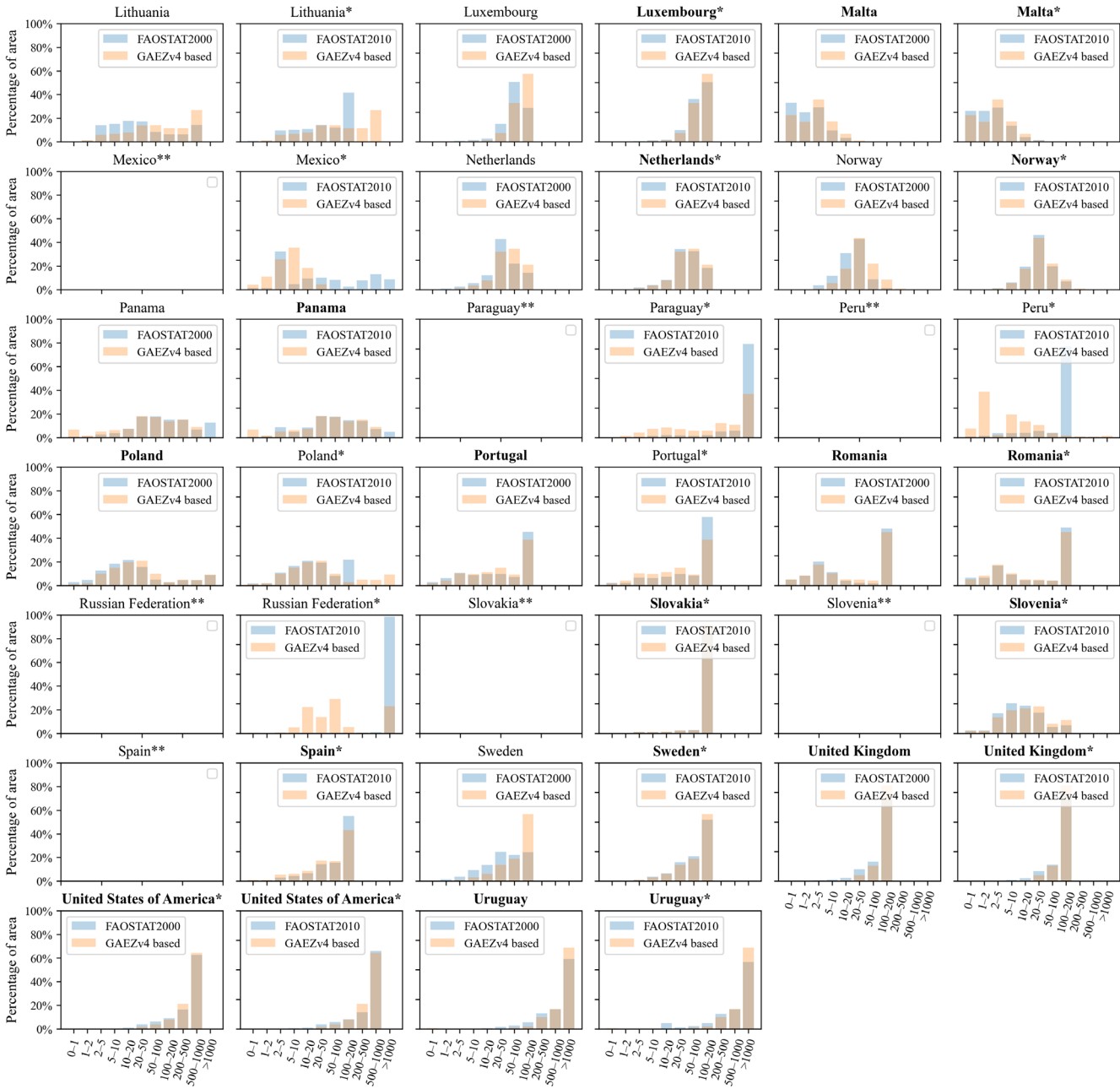

**Figure 7. Comparison of the percentage of total crop area operated by each farm size (non-crop-specific farm size structure) between FAOSTAT structural data for the year 2000 and 2010 (FAO, 2022) and our farm-size- and crop-specific dataset based on the GAEZv4 crop map. Bold font country titles indicate that farm size structures in FAOSTAT are similar to our dataset. Note that for the year 2000, farm size structure from FAOSTAT structural data is the same with Lowder et al. (2016) except for one country [S5]. Only the countries covered by our dataset and FAOSTAT are shown. The alternative version based on SPAM2010 crop map is given**
**in Fig. A3. * FAOSTAT provides (part of) the structural data by interpolating other reported data, not directly from countries' official reports. ** FAOSTAT provides no farm size structural data of the year 2000 or 2010 for comparison.**

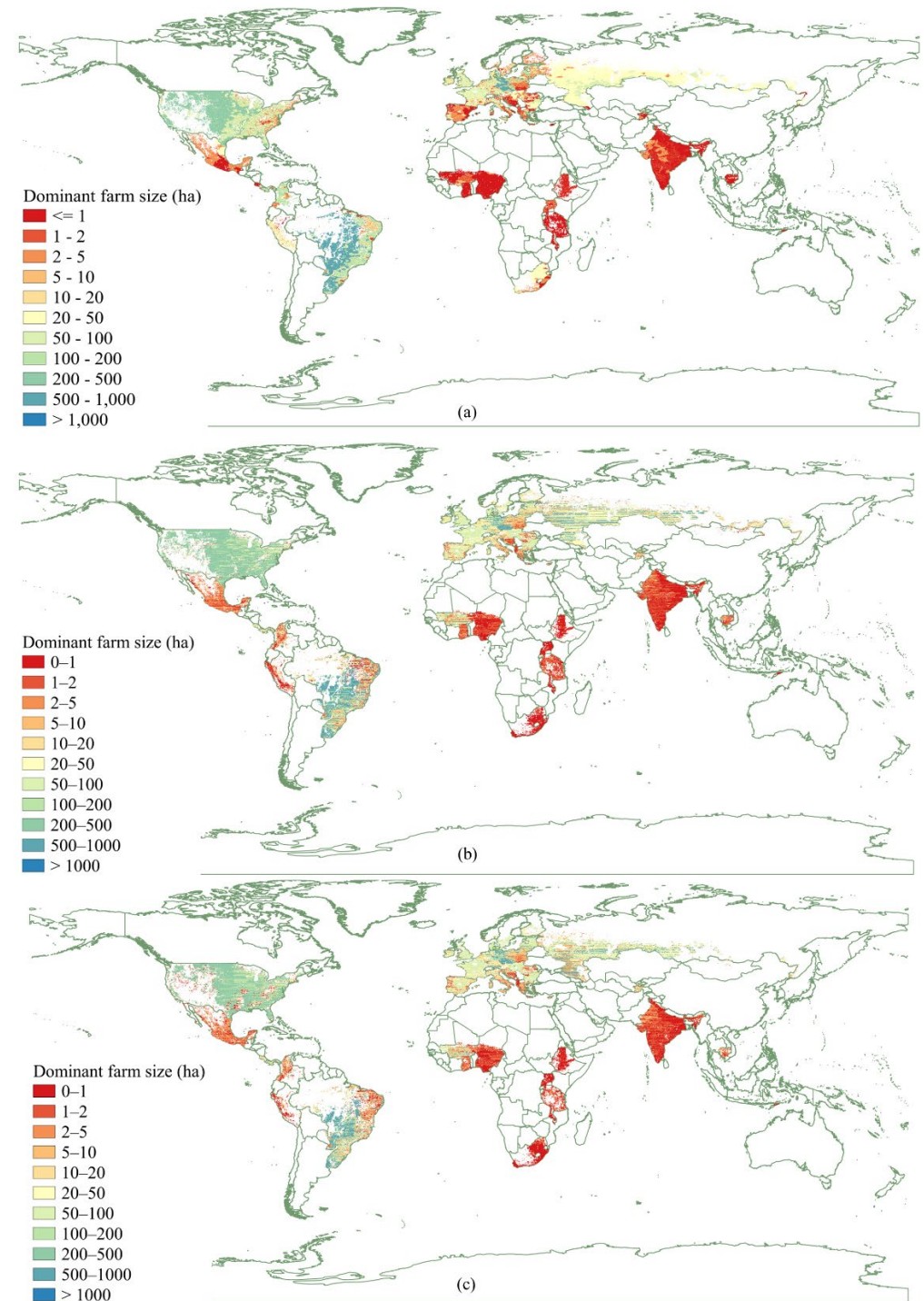

**Figure 8. Dominant farm size according to the dataset by Mehrabi et al. (2020) (a), our farm-size- and crop-specific dataset based on the GAEZv4 crop map (b) and SPAM2010 crop map (c), respectively. Only cells included in both the dataset by Mehrabi et al. (2020) and our datasets are shown.**

# 4 Discussion

## 4.1 Potential explanations for irrigation and farm size

Our datasets confirm findings by previous studies that smaller farms have a higher relative irrigation share compared to larger farms. This seems to be the case because relatively many of the small farms are located in severe water scarce regions, which would require them to irrigate more and more often to grow their crops (Fig. 9). However, it remains unclear whether small farms adapt to water scarcity via irrigation or that irrigation practices of small farms increase water scarcity (Grafton et al., 2018). Another explanation relates to the farm size structures between countries. Asian countries are home to the majority of small farms, and previous studies have shown that, on average, the relative share of irrigated area on Asian small farms is indeed much higher than in other countries, regardless regional water scarcity levels (Ricciardi et al., 2020).

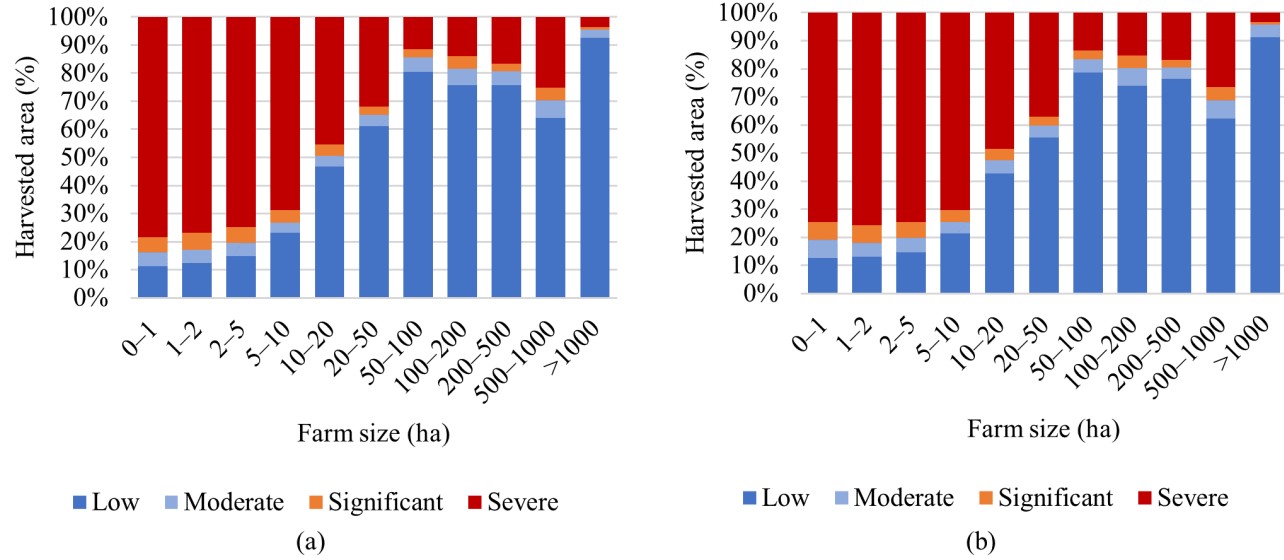

(a)                                                                 (b)

**Figure 9. Blue water scarcity levels (WSL) within each farm size according to our farm-size- and crop-specific harvested area dataset based on the GAEZv4 crop map (a) and the SPAM2010 crop map (b) under four blue water scarcity levels (WSL) by Mekonnen and Hoekstra (2016). Low blue WSL indicates blue water consumption does not exceed blue water availability; moderate WSL indicates blue water consumption is 100–150% of blue water availability; significant WSL indicates blue water consumption is 150–200% of blue water availability; and severe WSL indicates blue water consumption is larger than 200% of blue water availability.**

The irrigation of >1000 ha farm size shown by our datasets is relatively low, which could be explained by the regional climate and crop characteristics. Sugarcane in São Paulo, Brazil, is one of the main contributors to the significant and severe water scarce area of >1000 ha farm size. In these regions, water scarcity is not present all year round. The level of water scarcity is low from January to June, which is the tillering phase for sugarcane. Sugarcane is usually harvested during the dry season, desirably, during which moisture in sugarcane is relatively low and the sugar is highly concentrated (Kavats et al., 2020). This may help to explain why the large farms in this area are rainfed even though under a certain level of water scarcity.

## 4.2 Uncertainties

We hypothesized that uncertainties in crop maps might propagate to and influence uncertainties in our gridded datasets. Therefore, we developed two gridded datasets based on two different crop maps, i.e., GAEZv4 and SPAM2010. From the results and validation, we observed some differences in crop distribution, especially at the grid cell level. These differences reflect uncertainties in farmland location and affected the spatial validations on both farm-size-specific oil palm and dominant farm size distributions. At the same time, these uncertainties at grid cell level have a limited impact on country level statistics and validation, as can be seen in Fig. 3, Fig. 5, Fig. 6, and Fig. A2.

Differences—and therefore uncertainties—related to farming systems are more pronounced between the two crop maps, also at country level. Fig. 6 and [S8] show that our SPAM2010 based dataset yields lower irrigation ratios than that based on GAEZv4. This is likely the consequence of SPAM2010 defining irrigation as the actually irrigated area, whereas GAEZv4 defines irrigation by the area that is equipped with fully irrigation facilities. Despite these differences, however, findings of the overall relative irrigation share being higher of smaller farms and higher absolute irrigation of larger farms under elevated levels of water scarcity are supported by our datasets based on both crop maps.

The uncertainties in the crop maps also affect how we downscaled the dataset by Ricciardi et al. (2018a, b), the core source of our datasets. It occurred that crops could be found in the dataset by Ricciardi et al. (2018a, b) for a given administrative unit but not in the crop maps, or vice versa. The consequence of these inconsistencies was that 23.3% and 21.6% of the crop area in the dataset by Ricciardi et al. (2018a, b) could not be downscaled, respectively because the GAEZv4 or the SPAM2010 crop map indicated no crops were grown in those locations. Vice versa, 17.8% and 12.4% of the harvested area in the GAEZv4 and SPAM2010 crop maps, respectively, could not directly be assigned a farm size due to absent records in the dataset by Ricciardi et al. (2018a, b). Although these are substantial percentages of crop areas, our validation did not detect any peculiarities in outcomes attributable to these inconsistencies. Developing more accurate crop maps should reduce a substantial bit of the abovementioned uncertainties in the future.

Beside uncertainties propagated from input data, new uncertainties are introduced through our pre-processing procedures. In estimating crop-specific farm size structures using the dataset by Ricciardi et al. (2018a, b), ~12% of our final estimates were based on crop production instead of crop area. According to Ricciardi et al. (2018a, b), the introduced uncertainties are limited when using crop production. In addition, the year of the source data of Ricciardi et al. (2018a, b) ranges from 2001 to 2015 with median year of 2013, the transient nature of farm sizes, particularly in developing countries, may not be captured when it is used for the year of 2010.

The way we defined and apply constraints during the optimization process also introduced new uncertainties. Solving for the GAEZv4 and the SPAM2010 based datasets, we performed 7381 and 6017 optimizations, respectively. Differences in total number of optimizations can be explained by differences in cropland extent underlying both crop maps. Of their total number of optimizations, 4378 and 3671 needed to be relaxed using an elastic factor of 0.125 or smaller (Eq. (5)), for the respective crop maps, while 239 and 203 needed to be further relaxed by removing some of the minimum area constraints (Eq. (6) – (9)).

The latter relaxation of minimum area constraints introduced inconsistencies with the source dominant field size distribution, which further adds uncertainties to our datasets. This affected ~3% of our total calculations.

     In the optimization process, it further occurred that crop area needed to be allocated to a farm size that was not included in the dataset by Ricciardi et al. (2018a, b). This happened in cases where both the crop and part of the eleven farm sizes were included in the dataset by Ricciardi et al. (2018a, b), yet meeting the minimum area constraints required introducing an

additional farm size for the crop at hand. In such cases, we still ensured the 10% maximum relative difference with the dataset by Ricciardi et al. (2018a, b) to ensure the overall farm size structures. This uncertainty was introduced for ~0.1% and 5.0% of harvested area for the GAEZv4 and SPAM2010 based farm-size- and crop-specific datasets, respectively.

     Finally, despite the uncertainties at the grid cell level, the used datasets and our datasets were found to be more reliable at the country level. For example, the two crop maps were developed by downscaling the agriculture census at the (sub)national level.

Collected agriculture census and social-ecological factors considered during downscaling may lead to some differences at the grid cell level in the two crop maps, while they were all adjusted to the country level data from FAOSTAT. The dominant field size distribution is also uncertain at the grid cell level which was estimated by spatial interpolating of training samples. The uncertainty will decrease when the focus is on the regional level (Lesiv et al., 2019). Validations also show well consistencies with country level observations. Using GAEZv4 based and SPAM2010 based datasets at the same time helps to reduce

uncertainties at the grid cell level.

## 4.3 Limitations

     With the ambition to map simultaneously farm-size- and crop-specific harvested area, we were able to cover 56 countries based on state-of-the-art recent datasets (e.g. Ricciardi et al. (2018a, b), Lesiv et al. (2019), and Kim et al. (2021)). Although these countries reflect about half of the global cropland, the remaining countries could not be included due to lacking data

availability. Particularly farm-size-specific data is scarce or not publicly available for most of the excluded countries, but across-the-board data availability is the main obstacle in creating a dataset with global coverage. Approaches based on deep learning and remote sensing, similar to what Descals et al. (2020) did to obtain their oil palm dataset with which we validated some of our findings, may prove promising alternatives to mapping the global farm-size- and crop-specific harvested. However, the lack of farm size training samples and the enormous computational requirements are still challenges for such approaches

(Descals et al., 2020).

     Our estimations are based on planted crop and harvested area that is representative of the year 2010. Farmers' choice of crop will change along with climate, market demands, and many other factors. While our gridded datasets provide a robust baseline, it would be insightful to describe developments over time. However, capturing dynamics of harvested area under changing conditions and environments, particularly dynamic in developing countries (Giller et al., 2021), requires even more additional

data. Still, our datasets may be updated in the future for additional years, since many of the underlying datasets, including GAEZ, SPAM, and the cropland extent map by Latham et al. (2014) and Lu et al. (2020) are planned to be regularly updated. The dominant field size distribution by Lesiv et al. (2019) has already been updated since its first publication and announced

more updates in the future. Ricciardi et al. (2018a, b) did not share plans to update their dataset (yet), but it could be done using particularly data from the World Programme for the Census of Agriculture (FAO, 2015) and EUROSTAT ( EUROSTAT, 2021). We developed our model and code such that any updates and extensions of in the future are relatively easily incorporated.

## 4.4 Suggestions on developing farm-size- and crop-specific production dataset

Crop production of small farms is one of the main concerns of the Target 2.3 (double the agricultural productivity and the incomes of small-scale food producers) of SDG 2 (Zero hunger) (UNSD, 2022). It would therefore be a major achievement if we could develop farm-size-specific agricultural production dataset in support of this Target. However, compared to harvested areas, an empirical farm-size-specific dataset on production or yield is even more scarce. Thus, developing a farm-size- and crop-specific production dataset requires additional modeling and our datasets could readily be used as input for such development.

Developing a farm-size- and crop-specific production dataset requires unpacking the various factors that impact yield and are known or expected to correlate with farm size as recent studies show that the relationship between farm size and crop production is indirect and complex, cf. Muyanga and Jayne (2019) and Iizumi et al. (2021). Some factors could be unpacked directly for farm sizes with our datasets. For example, one could overlap our datasets with the soil and climate datasets to estimate soil and climate production conditions for each farm size. Other factors could be unpacked indirectly via agricultural production system, e.g. agricultural management and input factors. Specifying agricultural management and input factors according to farming systems could help to first evaluate crop yield for different farming systems, and then allocate the yield back to farm sizes according to farm size structure in each farming system. With unpacked factors, one could estimate the farm-size- and crop-specific production with our harvested area as input using crop models as well as GAEZv4 and SPAM2010.

## 5 Code and data availability

The code, source data, and resulting farm-size- and crop-specific harvested area datasets are freely available via a Creative Commons Attribution 4.0 International license at https://doi.org/10.5281/zenodo.6976249 (Su et al., 2022). The resulting datasets are available in *.csv and *.nc (netCDF) for each crop and farming system. For each crop, farming system, and farm size, we provide gridded harvested area in the coordinate Systems of EPSG:4326 - WGS 84. Gridded summaries over crops and farming systems are also available.

## 6 Conclusions

This study presents 5-arcmin gridded simultaneously farm-size- and crop-specific datasets of harvested area for 56 countries. The datasets are based on various state-of-the-art and recent datasets on farm-size- and/or crop-specific land use, cropland

extent, and dominant field size distribution. The resulting datasets show strong consistency along multiple variables validated against multiple empirical and published sources. While our high-resolution dataset fills a part of the data gap, lacking data availability is still hampering the development of dynamic datasets with full global coverage. Nevertheless, we are confident

that our current datasets will prove to be a useful tool for improving our understanding of differences between small- and large-scale farms, e.g. in terms of climate change adaptation and mitigation strategies, water consumption patterns, and contribution to (local) food security and SDG 2.

**Appendices**

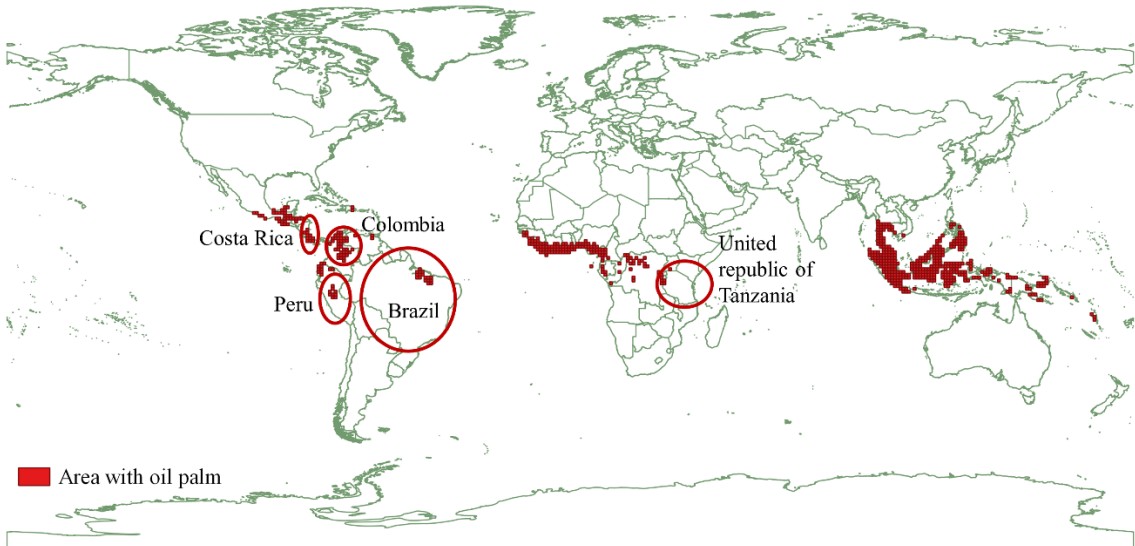

**Figure A1. The global distribution of oil palm according to Descals et al. (2020). The five countries for which we validated our datasets are circled in red.**

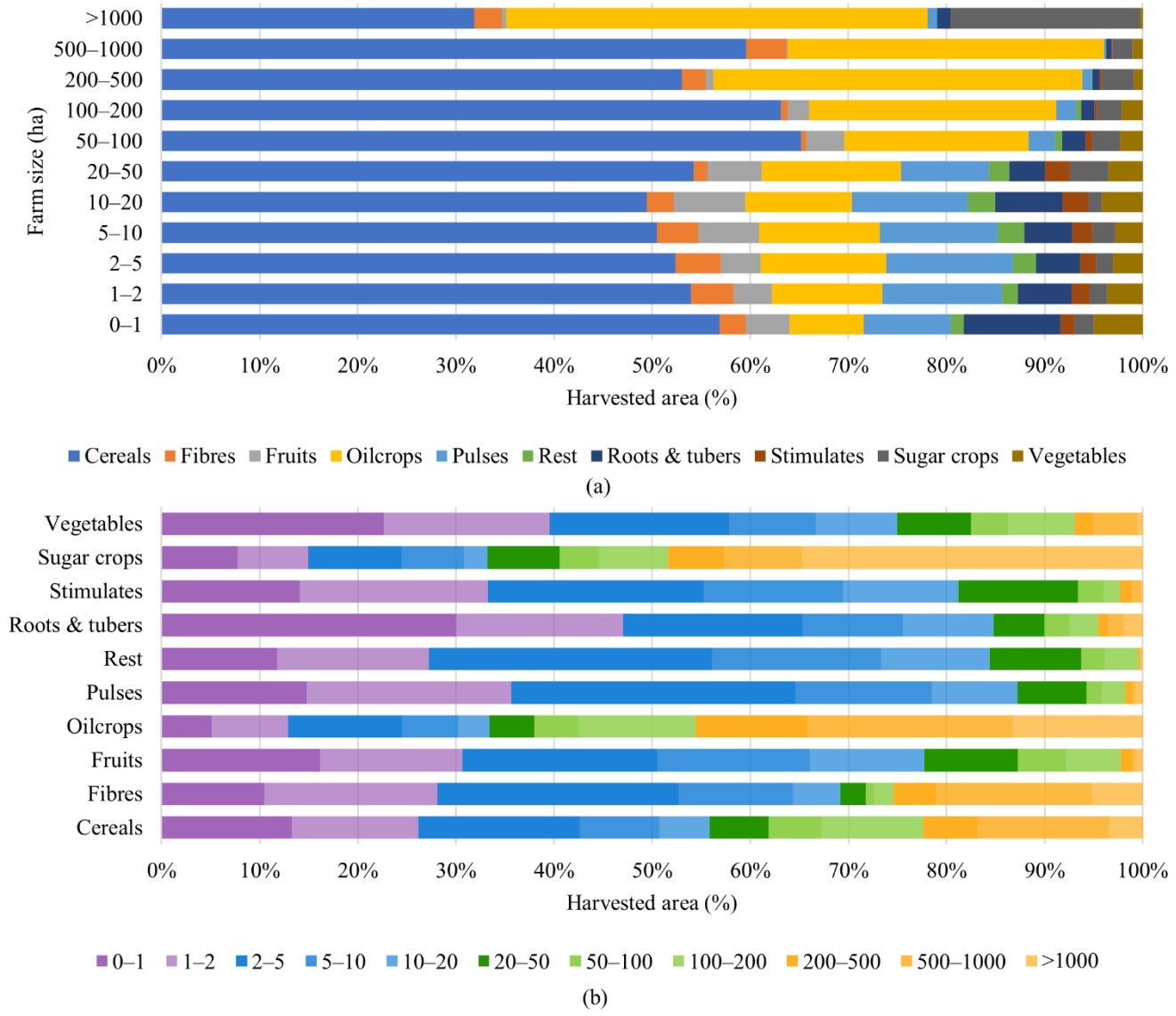

**Figure A2. Harvested area of crop groups within each farm size (a) and harvested area of crop groups by farm size (b) according to our farm-size- and crop-specific harvested area dataset based on the SPAM2010 crop map. The alternative version based on GAEZv4 crop map is given in Fig. 3.**

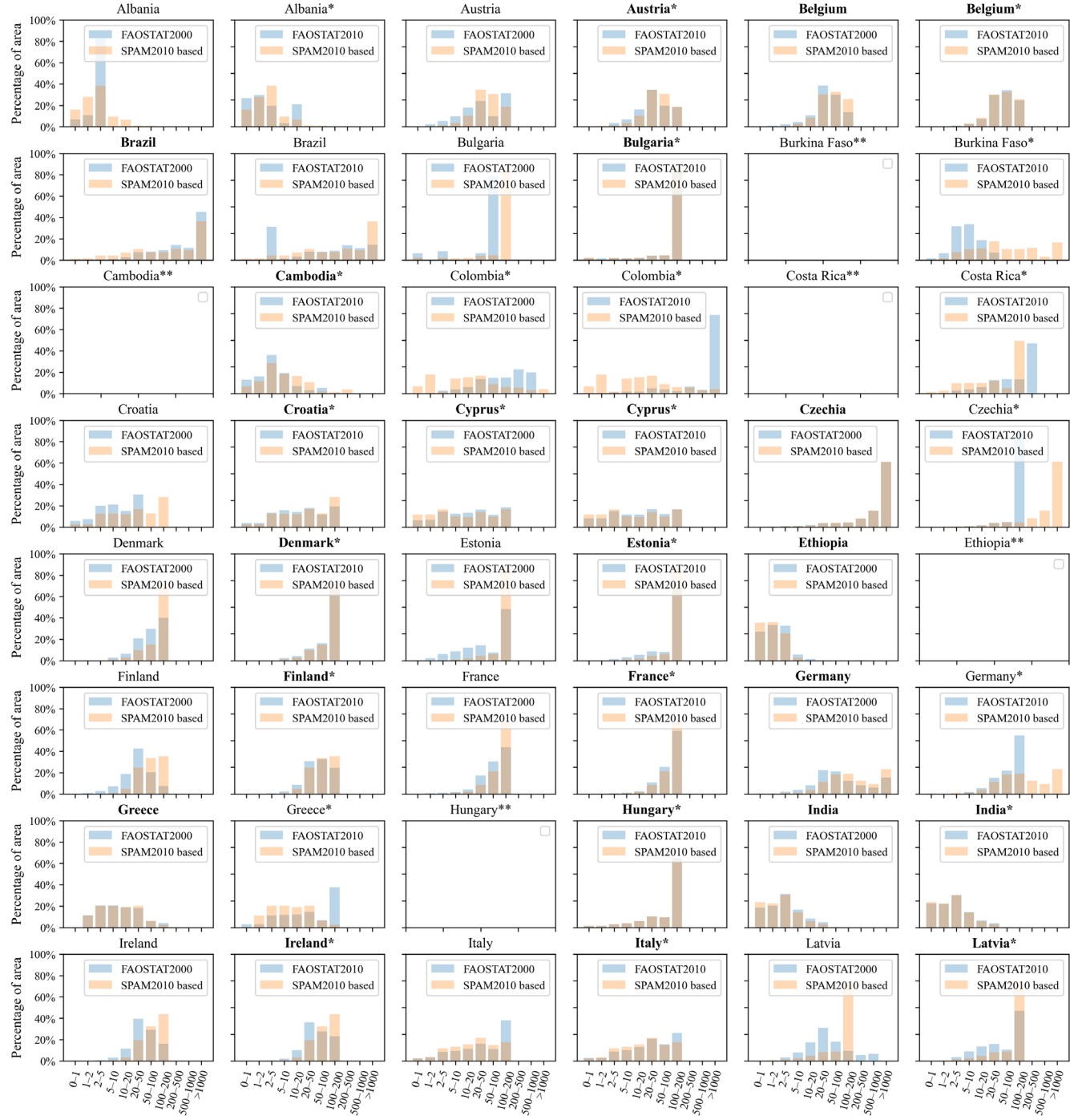

(Continued)

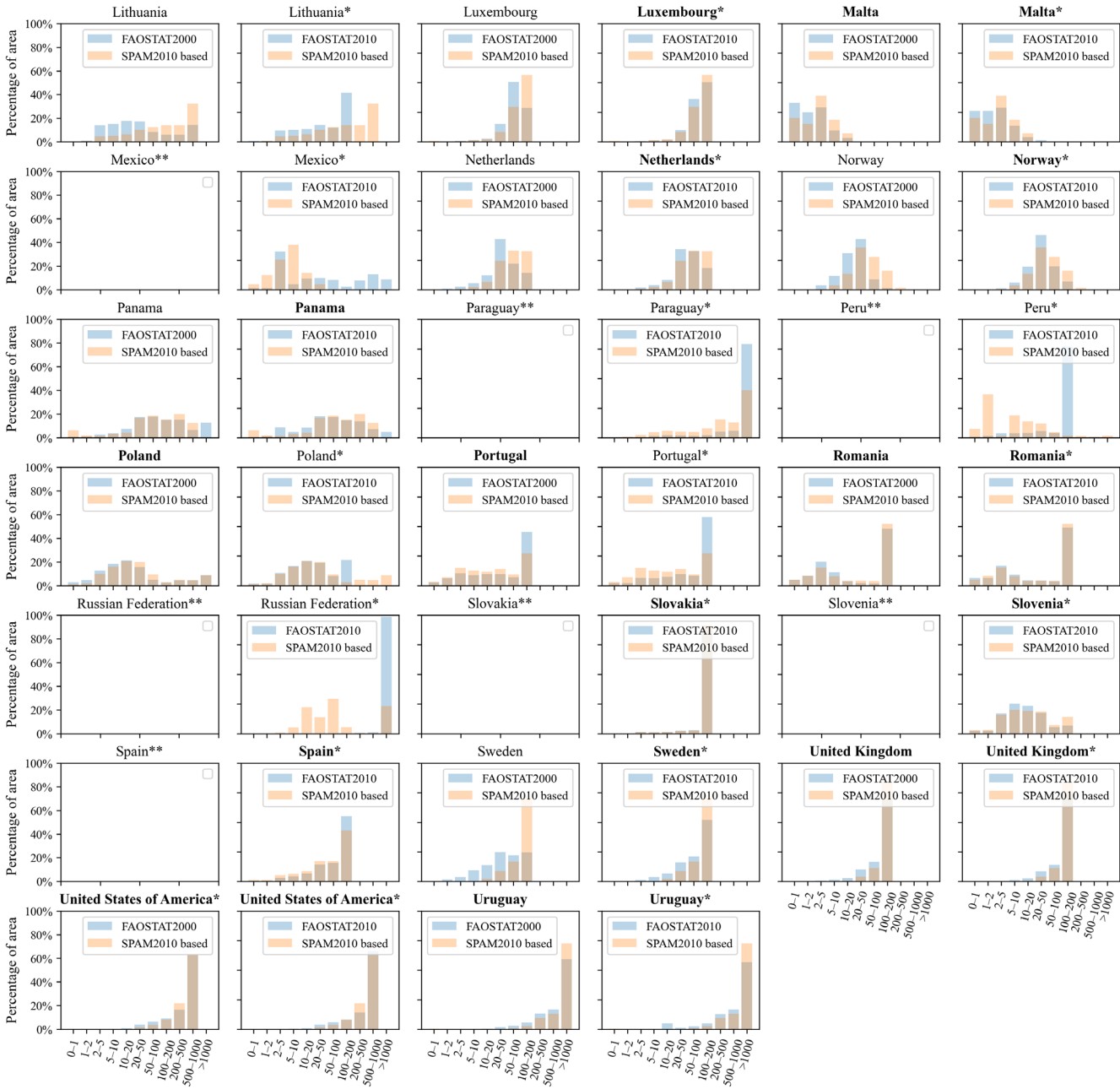

**Figure A3.** Comparison of the percentage of total crop area operated by each farm size (non-crop-specific farm size structure) between FAOSTAT structural data for the year 2000 and 2010 (FAO, 2022) and our farm-size- and crop-specific dataset based on the SPAM2010 crop map. Bold font country titles indicate that farm size structures in FAOSTAT are similar to our dataset. Note that for the year 2000, farm size structure from FAOSTAT structural data is the same with Lowder et al. (2016) except for one country [S5]. Only the countries covered by our dataset and FAOSTAT are shown. The alternative version based on GAEZv4 crop map is given in Fig. 7. * FAOSTAT provides (part of) the structural data by interpolating other reported data, not directly from countries' official reports. ** FAOSTAT provides no farm size structural data of the year 2000 or 2010 for comparison.

## Author contribution

The concept of this work originated from HS in collaboration with MSK and RJH. The study was designed and conducted by HS under the supervision of BM and DLG with feedback from MSK and RJH. HS wrote the draft of this manuscript. HS, BM, DLG, MSK, and RJH participated in the analysis of results and revision and editing of the manuscript.

## Competing interests

The authors declare that they have no conflict of interest.

## Acknowledgements

This study is funded in part by the European Research Council (ERC) Advanced Grant 2018 (action number 834716) and the University of Twente. Part of the research was developed in the Young Scientists Summer Program (YSSP) at the International Institute for Applied Systems Analysis, Laxenburg (Austria).

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
