# Peer review of "Gridded 5-arcmin, simultaneously farm-size- and crop-specific harvested area for 56 countries"

_Earth System Science Data, 2022_

## Referee Comment (RC1)

Comments to the manuscript entitled "Gridded 5-arcmin, simultaneously farm-size- and crop-specific harvested area for 56 countries" by Han Su et al. (essd-2022-72)

General comment:
I respect the authors' challenge in compiling farm-size- and crop-specific harvested area datasets like one presented in this study. Although there might be much room for further validation of the developed dataset, I would not request it since it is in realty difficult to objectively assess the uncertainties of the dataset when no similar dataset is available. My comments are mostly from editorial point of view, and to improve the manuscript further.

Relatively major comments:
1. I did not find any list of the 56 countries covered in this dataset. Probably, Table S6 is close to the list, but it might be incomplete in the case that Meharabi's dataset and your dataset are not overlapped. Related to this, why don't you present your dataset as the map in main text for demonstration purpose? Showing a map of main variable of your dataset is help readers understand your dataset.
2. Since some farming types (e.g., the rainfed subsistence in SPAM2010) are assumed to be an indicator of small-scale farmers in literature (e.g., Iizumi et al., 2021), it would be great if you could show how the individual farming type considered here correlate with field size or not, using your dataset.
3. I would encourage the authors to add a brief discussion towards next step – specifically, compiling a farm-size and crop-specific production or yield dataset. Increasing productivity of small-scale farmers is a main goal in SDG 2 (zero hanger). Once farm-size harvested area datasets like one presented hare become available, then people expect farm-size- and crop-specific yield dataset to calculate the production share of small-scale farmers. But it is elusive how yield differ by farm size (e.g., Muyanga and Jayne (2019) and Supplementary Text of Iizumi et al. (2021)). What is your though on the current feasibility and limitations to develop such dataset?

Specific comments:
4. Table1. The units of spatial resolution are mixed (arcmin and km). Using a consistent unit or showing an indication for conversion (for instance, approximately 10 km for 5 arcmin) increase readability.
5. L129. Can you add a brief definition of crop area, planted area, harvested area and cultivated area? Especially, are crop area and cultivated area used here crop-specific?
6. L164. "the total harvested" -> "the total area harvested"
7. L216. "access" -> "assess"
8. Fig. 3. How did you associate farm size with the water scarcity levels of Hoekstra et al. (2012)? Since the water scarcity level data are on monthly resolution, did you calculate an average for cropping season?
9. L308-309. This is rather speculative. At least, relevant citations are needed to support this statement on change in farm size for ten-year period. And for your reference, in their Fig. 2, Yu et al. (2013) reports based on farmer interview that change in farmland area per household increase from 1.3 ha in the early 1980s to 2.6 he in the early 2010s for some villages in North China. Although you have talked here about Bulgaria, which could be largely different with China, it seems that the difference (78.5% and

5% of harvested area is under the farm size 50-100 ha in Lowder's dataset and your dataset, respectively) is too large to be explain by the difference in the reported time.

10. L364. I think the social-ecological factors mentioned here indicate the use of GAEZ. Although this reasoning may be true, there is no result to show what social-ecological factors lead to the difference in the two crop maps.

References
· Iizumi, T. et al. (2021) Soil carbon-food synergy: sizable contributions of small-scale farmers. CABI Agric Biosci 2, 43. https://doi.org/10.1186/s43170-021-00063-6
· Muyanga, M. and Jayne, T.S. (2019) Revisiting the farm size-productivity relationship based on a relatively wide range of farm sizes: Evidence from Kenya. Amer J Agric Econ. 101, 1140–1163. https://doi.org/10.1093/ajae/aaz003
· Yu, Q. et al. (2013) A survey-based exploration of land-system dynamics in an agricultural region of Northeast China. Agricultural Systems, 121, 106-116. https://doi.org/10.1016/j.agsy.2013.06.006

---

## Author Response (AR1)

Dear Dr. Hanqin Tian,

We would like to thank you and the reviewers for the time and effort spent on reviewing. All the comments are fully considered during the revision. Below you could find our responses, structured as: **[Comment]** from referees, [Response] from authors, and [Change] made in the manuscript (clean mode and track mode).

Kind regards,

Han Su, on behalf of all co-authors

PhD Candidate

Multidisciplinary Water Management group, University of Twente

**Referee #1 (RC1, RC2, and RC3)**

- ## RC1

**General comment:**
**[Comment] I respect the authors' challenge in compiling farm-size- and crop-specific harvested area datasets like one presented in this study. Although there might be much room for further validation of the developed dataset, I would not request it since it is in realty difficult to objectively assess the uncertainties of the dataset when no similar dataset is available. My comments are mostly from editorial point of view, and to improve the manuscript further.**

[Response] We would like to thank you for the time and effort spent on reviewing. We appreciate your comments which enable us to improve our manuscript. We provide our responses below.

**Relatively major comments:**
**[Comment] 1. I did not find any list of the 56 countries covered in this dataset. Probably, Table S6 is close to the list, but it might be incomplete in the case that Meharabi's dataset and your dataset are not overlapped. Related to this, why don't you present your dataset as the map in main text for demonstration purpose? Showing a map of main variable of your dataset is help readers understand your dataset.**

[Response] We agree that the complete list of the 56 countries, taken from Ricciardi's dataset, is missing in the manuscript, and will add it as supplementary materials. We will also add maps on the harvested area of rainfed maize belonging to two farm sizes (2-5 ha and 500-1000 ha) in the next revision to illustrate some of the multiple dimensions of the dataset in a limited number of maps.

The list of 56 countries:

Table. The list of 56 countries and country code

| No. | Country code | Country | No. | Country code | Country |
|---|---|---|---|---|---|
| 1 | ALB | Albania | 29 | LUX | Luxembourg |
| 2 | AUT | Austria | 30 | LVA | Latvia |
| 3 | BEL | Belgium | 31 | MEX | Mexico |
| 4 | BFA | Burkina Faso | 32 | MLI | Mali |
| 5 | BGR | Bulgaria | 33 | MLT | Malta |
| 6 | BIH | Bosnia and Herzegovina | 34 | MNG | Mongolia |
| 7 | BRA | Brazil | 35 | MWI | Malawi |
| 8 | COL | Colombia | 36 | NER | Niger |
| 9 | COS | Costa Rica | 37 | NGA | Nigeria |
| 10 | CYP | Cyprus | 38 | NLD | Netherlands |
| 11 | CZE | Czechia | 39 | NOR | Norway |
| 12 | DEU | Germany | 40 | PAN | Panama |
| 13 | DNK | Denmark | 41 | PER | Peru |
| 14 | ESP | Spain | 42 | POL | Poland |
| 15 | EST | Estonia | 43 | PRT | Portugal |
| 16 | ETH | Ethiopia | 44 | PRY | Paraguay |
| 17 | FIN | Finland | 45 | ROM | Romania |
| 18 | FRA | France | 46 | RUS | Russian Federation |
| 19 | GBR | United Kingdom | 47 | SVK | Slovakia |
| 20 | GHA | Ghana | 48 | SVN | Slovenia |
| 21 | GRC | Greece | 49 | SWE | Sweden |
| 22 | HRV | Croatia | 50 | TJK | Tajikistan |
| 23 | HUN | Hungary | 51 | TLS | Timor-Leste |
| 24 | IND | India | 52 | TZA | United Republic of Tanzania |
| 25 | IRL | Ireland | 53 | UGA | Uganda |
| 26 | ITA | Italy | 54 | URY | Uruguay |
| 27 | KHM | Cambodia | 55 | USA | United States of America |
| 28 | LTU | Lithuania | 56 | ZAF | South Africa |

The maps for demonstration purposes:

[Figure]

Fig. The gird cells with a harvested area of rainfed maize belonging to the farm size 2–5 ha (a) and farm size 500–1000 ha (b), according to the GAEZ based downscaled map.

[Change] The complete list of the 56 countries was add in the supplementary material, [S1]. The figure for illustration was added as Fig. 2, in section 3.1, and lines 251-253 (lines 257-259 in track mode).

**[Comment] 2. Since some farming types (e.g., the rainfed subsistence in SPAM2010) are assumed to be an indicator of small-scale farmers in literature (e.g., Iizumi et al. (2021)), it would be great if you could show how the individual farming type considered here correlate with field size or not, using your dataset.**

[Response] We agree that the combined data on farm size and farming type is valuable in providing insights into agriculture structure and is worthwhile to be illustrated in the paper. We will add the distribution of farming systems within each farm size in the next revision.

Our dataset indicates the subsistence and low-input rainfed farming system is mainly operated at smaller farms, but the smaller farms do not exclusively consist of subsistence and low-input rainfed farming system: they also operate a significant portion of the irrigated and high-input rainfed area. Similarly, the main type of farming system of larger farms is high-input rainfed, but the high-input rainfed is far from being limited to larger farms.

[Figure]

Fig. The distribution of irrigated, low- and high-input rainfed, and subsistence rainfed farming systems within each farm size according to the SPAM based downscaled map

[Change] Illustration on farming systems was added as Fig. 4, in section 3.2, and lines 299-303 (lines 310-314 in track mode).

**[Comment] 3. I would encourage the authors to add a brief discussion towards next step – specifically, compiling a farm-size and crop-specific production or yield dataset. Increasing productivity of small-scale farmers is a main goal in SDG 2 (zero hanger). Once farm-size harvested area datasets like one presented hare become available, then people expect farm-size- and crop-specific yield dataset to calculate the production share of small-scale farmers. But it is elusive how yield differ by farm size (e.g., Muyanga and Jayne (2019) and Supplementary Text of Iizumi et al. (2021)). What is your though on the current feasibility and limitations to develop such dataset?**

[Response] One of the underlying aims of constructing the current dataset is to compile the best-available empirical farm-size specific dataset. Compared to harvested area, an empirical farm-size specific dataset on yield or production is even more scarce. The data on yield or production of farm sizes is available for limited countries, but those countries are not always the most vulnerable in terms of food insecurity. Developing farm-size specific maps on yield or production may be the goal of further research and may be one of the applications of our dataset that directly benefit from the additional dimensionality achieved. Such datasets would require estimating the yield based on additional datasets or models.

As pointed out by the reviewer, correlations between farm size and yield are still under debate. Many factors contribute to this relationship, including but not limited to crop types, fertilizer input, climate, and soil conditions. The farm size itself does not directly affect yield, but farm size often correlates with factors that affect yield.

So, estimating crop yield for different farm sizes requires first unpacking the factors that directly impact yield and correlate with farm sizes. For environmental factors like soil conditions and climate, this could be achieved by overlapping our dataset with the soil and climate database. Agricultural management and input factors, like fertilizer input, could be

inferred from the agricultural production system data. Specifying agricultural management and input factors according to farming systems could help to first evaluate crop yield for different farming systems, and then allocate the yield back to farm sizes according to their proportion in each farming system. Such an approach would rely on the assumption that agricultural management practices of different farming systems do not depend on farm size. Reliable estimations of yield for different farming systems could be either derived from SPAM2010 and GAEZ v4 data or based on crop modeling when the data on the factors are available.

We will add the above discussion in the next revision.

[Change] The above discussion on farm-size-specific production was added as section 4.3, lines 434-451 (lines 446-463 in track mode).

**Specific comments:**
**[Comment] 4. Table1. The units of spatial resolution are mixed (arcmin and km). Using a consistent unit or showing an indication for conversion (for instance, approximately 10 km for 5 arcmin) increase readability.**

[Response] We agree with your suggestion. We will add indications for unit conversion.

[Change] The unit conversion was added in Table 1, lines 125-126 (lines 129-130 in track mode).

**[Comment] 5. L129. Can you add a brief definition of crop area, planted area, harvested area and cultivated area? Especially, are crop area and cultivated area used here crop-specific?**

[Response] The crop area, planted area, harvested area, and cultivated area is crop-specific. These variables were identified by Ricciardi's dataset from the local agriculture census. There is no worldwide standard definition for these items (FAO, 2015). Local agriculture censuses have their preference to use one of them for specific crops. Generally speaking, planted area is used for temporary crops; cultivated area for temporary crops and permanent crops; crop area for temporary crops, permanent crops, fallow, meadows, and pastures; harvested area is the cultivated area excluding the area destroyed by natural disasters or other reasons (FAO, 2015, 2020). We will clarify them in the next revision.

[Change] We clarified these items in section 2.2, lines 130-135 (135-140).

**[Comment] 6. L164. "the total harvested" -> "the total area harvested"**

[Response] We agree with your suggestion. This phrase will be corrected.

[Change] We fixed this phrase, section 2.3, line 170 (line 175 in track mode).

**[Comment] 7. L216. "access" -> "assess"**

[Response] We agree with your suggestion. This word will be corrected.

[Change] We fixed this phrase, section 2.5, line 222 (line 227 in track mode).

**[Comment] 8. Fig. 3. How did you associate farm size with the water scarcity levels of Hoekstra et al. (2012)? Since the water scarcity level data are on monthly resolution, did you calculate an average for cropping season?**

[Response] We are sorry that there is a mistake in the reference here. We used the updated water scarcity map of Hoekstra et al. (2012), Mekonnen and Hoekstra (2016). In the updated dataset, there are monthly water scarcities and also an annual average of monthly blue water scarcity. We used the latter one. We will correct the reference and clarify the data source in the next revision.

[Change] We corrected the reference and clarified the data source, section 3.2, line 275 (line 281 in track mode).

**[Comment] 9. L308-309. This is rather speculative. At least, relevant citations are needed to support this statement on change in farm size for ten-year period. And for your reference, in their Fig. 2, Yu et al. (2013) reports based on farmer interview that change in farmland area per household increase from 1.3 ha in the early 1980s to 2.6 he in the early 2010s for some villages in North China. Although you have talked here about Bulgaria, which could be largely different with China, it seems that the difference (78.5% and 5% of harvested area is under the farm size 50-100 ha in Lowder's dataset and your dataset, respectively) is too large to be explain by the difference in the reported time.**

[Response] Thanks for pointing it out. Here, we want to emphasize both our results and other datasets indicate large farms are the major farm size in the country, but you are right, we also need to explain the difference better. How datasets process the farm size class may contribute to the differences besides the reported time. The farm size classes collected from the local agriculture census usually need to be harmonized into a classification system. Different datasets may have their own choice during this process. This may lead to the differences shown in the comparison, especially when the major farm sizes are similar but not the same.

We will add some explanations in the next revision.

[Change] We added the above explanation in the section 3.5, lines 350-354 (lines 363-365 in track mode).

**[Comment] 10. L364. I think the social-ecological factors mentioned here indicate the use of GAEZ. Although this reasoning may be true, there is no result to show what social-ecological factors lead to the difference in the two crop maps.**

[Response] Indeed, the social-ecological factors were considered in both GAEZ and SPAM. Quantifying how the use of different social-ecological factors may lead to differences in the two crop maps however is beyond the scope of this manuscript. Instead, we will weaken this statement in the next revision.

[Change] We weakened this statement in section 4.1, line 409 (line 421 in track mode).

- **RC2**

**[Comment] In relation to the author's response to [Comment] 2, I'm very much impressed by the figure (Fig. The distribution of irrigated, low- and high-input rainfed,**

and subsistence rainfed farming systems within each farm size according to the SPAM based downscaled map) that the portion of the subsistence rainfed and low-input rainfed farming systems account for more in the smaller farm sizes than in the larger farm sizes. The figure also shows that the portion of the irrigated farming system is more in the smaller farm sizes. Why is the irrigation-equipped area share relatively high in small size farmers? I would be appreciate it if the authors could explain this point. I suspect that this is due to the large number of small size farmers in Asia (in particular India) where water resources are abundant thanks to monsoon rainfalls.

[Response] Thank you for your comment. Indeed, a higher portion of irrigated farming system in smaller farms is shown in the figure you refer to, as well as in Fig. 3 in our manuscript this is supported by previous evidence (FAO, 2021; Ricciardi et al., 2020). The inclusion of the 56 countries and exclusion of other countries affect this estimation, but for the 56 countries, the overall higher portion of irrigated area in smaller farms correlates with the level of water scarcity: Fig. 3 in the manuscript indicates that higher portions of smaller farms are located in water-scarce regions as compared to larger farms. In the water-scarce regions, the percentage of the irrigated area could reach on average 40% for small farms. For India, the water scarcity map of Mekonnen and Hoekstra (2016) indicates a large part of India is under water scarcity from January to June, and thus under water scarcity on an annual average. The India agriculture input survey (DAFW, 2022) indicates 47.8% of the crop area belonging to farm size 0-2 ha was irrigated in India during 2011-2012. Thus, water scarcity may partly contribute to the high portion of irrigated areas in Indian small farms. Asian smaller farms also contribute to the higher irrigation portion in another way. In Asian countries including India, previous studies show that independent of regional water scarcity, on average the percentage of irrigated area in small farms is high: over 50% when water is scarce and over 20% when water is not scarce (Ricciardi et al., 2020). This percentage is much higher than that in Europe, Central Asia, Latin America, and Sub-Saharan Africa (Ricciardi et al., 2020). Since a large number of small farms are from Asia, the overall portion of irrigated areas in small farms is high. We will add the above analysis in the next revision.

[Change] We added the above additional explanation on the overall higher irrigation of smaller farms in section 3.2, lines 272-281 (lines 278-291 in track mode).

- ## RC3

[Comment] Thank you very much for your clarifications that is convincing. I look forward to see a revised manuscript.

[Response] Thank you very much.

**Referee #2 (RC4)**

- ## RC4

[Comment] This study tries to map the global distribution of farm size using data harmonization approach. This is an interesting topic, but there are a few major issues that need to be solved.

[Response] Thank you for your comments. These comments enable us to improve our manuscript. We appreciate the time and effort you spent on reviewing. Below are our responses and how we will address them in the next revision.

[Comment] **First, there is a large gap in China, causing an unpleasant blank area in the entire East Asia. I believe China's data can be easily obtained from the annual yearbook or other statistical records, and I would suggest the authors fill this gap.**

[Response] The inclusion of China is our ambition since designing the research, however, data access remains unsolved so far. To include any extra country or region, we need farm-size specific and crop-specific data at the regional level from statistical records. This information for China is not publicly available, which is confirmed by the *Statistics Information Service* from the *National Bureau of Statistics of China* after consulting. According to our best knowledge, two databases may provide such data: the microdata of the *Third National Agricultural Census in China* (NBS, 2022) and the *China Rural Household Panel Survey* (CRHPS) (SSECZU, 2019). We submitted our data request and discussed with the database manager of the two databases in August 2021 and February 2022 respectively, however, we could not be granted access according to the corresponding current data policy. The data policy might change in the future, and we are prepared to include more countries including China once additional data is available. We would also like to invite scholars, users, and policymakers to update our database together in the future.

[Change] We added a note on updating our database when additional data is available in section 4.2, lines 432-433 (lines 444-445 in track mode).

[Comment] **Second, I have concerns about the validation in Lines 220-224. The comparisons are actually a compromise of data inconsistency. What if a different threshold value was used? Do the conclusions change if a different threshold was used? A sensitivity analysis maybe helpful here.**

[Response] We agree that a sensitivity analysis would be helpful to understand the comparison here. Besides the current threshold of 25 ha, we also tried 10 ha and 50 ha as thresholds and conducted the same comparison with observations from satellite images. We found the conclusions in Section 3.3 are not sensitive to the choice of threshold. We will add the sensitivity analysis in the next revision.

[Change] We added the sensitivity analysis in section 3.3, lines 311-313 (lines 322-324 in track mode).

[Comment] **Third, language editing is also needed.**

[Response] The next revision will receive proofreading from a native speaker.

[Change] The manuscript was polished to improve the readability.

**Other minor suggestions:**
[Comment] **1. Line 119, an extra "and"?**

[Response] Yes, this word is redundant and will be removed in the next revision.

[Change] The redundant word was deleted, section 2.2, line 120 (line 124 in track mode).

**[Comment] 2. The claims in Lines 263-264 were actually not supported by the figure. There is a large drop in the >1000 category in Fig. 3a for the orange and red lines. Please also explain.**

[Response] Thanks for pointing it out; we agree that more precise formulation is due. The more appropriate claim will be that large farms irrigate to a larger extent than small farms when water is scarce.

The reason for the drop is that the water scarce area of the >1000 ha farm size is mainly contributed by limited crops from a few regions, at least in our dataset. In this case, the characteristics of these crops and regions have more impact on the overall relationship between water scarcity and irrigation. For example, one of the main contributors to the significant and severe water scarce area of >1000 ha farm size (the orange and red lines) is sugarcane from São Paulo in Brazil. Brazil is the world's largest sugarcane producer and São Paulo account for around 60% of sugarcane production in Brazil (Bordonal et al., 2018; Palludeto et al., 2018). Sugarcane in this area is dominated by >1000 ha farm size (Ricciardi et al., 2018), mainly rainfed (OECD-FAO, 2015; Yu et al., 2020), and under water scarcity (Mekonnen and Hoekstra, 2016). However, water scarcity is not present all year round. The level of water scarcity is low from January to June, which is the tillering phase for sugarcane. During the dry season, sugarcane is usually harvested, during which moisture in sugarcane is relatively low and the sugar is highly concentrated (Kavats et al., 2020). This may help to explain why the large farms in this area are rainfed even though under a certain level of water scarcity.

In Fig. 3, we do not aim to draw conclusions on irrigation levels for specific farm sizes in absence of further investigation on influencing factors and uncertainties. The reason we have Fig. 3 is to compare it with previous observations. Ricciardi et al. (2020) show that large farms irrigate to a larger extent than small farms when water is scarce. In their study, farms are divided into either small or large farms without further classification, and the status of water scarcity is only classified as the water is scarce (moderate, significant, and severe) or not (low). Plausible thresholds to differentiate small and large farms could be country specific, and range from 1-42 ha for most countries (FAO, 2017, 2019; Khalil et al., 2017). With any threshold within this range, our dataset supports previous observations given that the farm size >1000 ha only contributes to less than 4.5% of water scarce area of large farms, so specific observations for the largest farm size may be spurious and are not emphasized in the paper.

In the next revision, we will improve the claim, clarify the intention of this analysis, and explain Fig. 3 with more details based on the above response.

[Change] We improved the claim, clarified the intention of the analysis, and added explanations on the >1000 ha farm size in section 3.2, lines 267-271, 282-295 (lines 273-277, 293-306 in track mode).

**[Comment] 3. In line273, I don't know why the author made this claim: "This means the spatial distributions of oil palm production in our downscaled maps and Descals et al. (2020) are similar." The comparisons were about the harvested area, and why and how did the production involved here?**

[Response] Thanks for pointing it out. The statement indeed is about the harvested area instead of production. We will formulate it unambiguously in the next revision.

[Change] We clarified the claim in section 3.3, line 310 (lines 321-322 in track mode).

**[Comment] 4. Line 328, separately?**

[Response] Yes, this word will be corrected in the next revision.

[Change] The word was corrected in section 4.1, line 373 (line 384 in track mode).

**Reference**

Bordonal RO, Carvalho JLN, Lal R, de Figueiredo EB, de Oliveira BG, La Scala N (2018) Sustainability of sugarcane production in Brazil. A review. Agronomy for Sustainable Development 38. doi:10.1007/s13593-018-0490-x

DAFW (2022) Agriculture Census, Input Survey Dashboard, National Tables, All India tables, 2011-2012, Table 2A & 2B. Department of Agriculture & Farmers Welfare, Government of India

FAO (2015) World programme for the census of agriculture 2020, Volume 1, Programme, concepts and definitions. Rome

FAO (2017) Small family farms data portrait. Basic information document. Methodology and data description. Food and Agriculture Organization of the United Nations, Rome

FAO (2019) Methodology for computing and monitoring the Sustainable Development Goal indicators 2.3.1 and 2.3.2. FAO Statistics Working Paper Series 18-14. Food and Agriculture Organization of the United Nations, Rome

FAO (2020) RuLIS Codebook, Rural Livelihoods Information System. Rome

FAO (2021) RuLIS - Rural Livelihoods Information System.

Hoekstra AY, Mekonnen MM, Chapagain AK, Mathews RE, Richter BD (2012) Global monthly water scarcity: blue water footprints versus blue water availability. PLoS One 7:e32688. doi:10.1371/journal.pone.0032688

Iizumi T, Hosokawa N, Wagai R (2021) Soil carbon-food synergy: sizable contributions of small-scale farmers. CABI Agriculture and Bioscience 2:43. doi:10.1186/s43170-021-00063-6

Kavats O, Khramov D, Sergieieva K, Vasyliev V (2020) Monitoring of sugarcane harvest in Brazil based on optical and SAR data. Remote Sensing 12:1-26. doi:10.3390/rs12244080

Khalil CA, Conforti P, Ergin I, Gennari P (2017) Defining small scale food producers to monitor target 2.3 of the 2030 Agenda for Sustainable Development. FAO, Rome

Mekonnen MM, Hoekstra AY (2016) Four billion people facing severe water scarcity. Science Advances 2. doi:10.1126/sciadv.1500323

Muyanga M, Jayne TS (2019) Revisiting the Farm Size-Productivity Relationship Based on a Relatively Wide Range of Farm Sizes: Evidence from Kenya. 101:1140-1163. doi:https://doi.org/10.1093/ajae/aaz003

NBS (2022) Micro data, National Bureau of Statistics. https://microdata.stats.gov.cn/ (in Chinese) Accessed 20-April-2022

OECD-FAO (2015) OECD-FAO Agricultural Outlook 2015-2024. Organisation for Economic Co-operation Development, Food and Agriculture Organization of the United Nations

Palludeto AWA, Telles TS, Souza RF, de Moura FR (2018) Sugarcane expansion and farmland prices in São Paulo State, Brazil. Agriculture and Food Security 7. doi:10.1186/s40066-017-0141-5

Ricciardi V, Ramankutty N, Mehrabi Z, Jarvis L, Chookolingo B (2018) An open-access dataset of crop production by farm size from agricultural censuses and surveys. Data Brief 19:1970-1988. doi:10.1016/j.dib.2018.06.057

Ricciardi V, Wane A, Sidhu BS, Godde C, Solomon D, McCullough E, Diekmann F, Porciello J, Jain M, Randall N (2020) A scoping review of research funding for small-scale farmers in water scarce regions. Nature Sustainability 3:836-844.

SSECZU (2019) Data access policy of the Chinese Family Database from Zhejiang University. http://ssec.zju.edu.cn/sites/main/template/news.aspx?id=51027 (in Chinese) Accessed 20-April-2022

Yu Q, Wu W, Verburg PH, van Vliet J, Yang P, Zhou Q, Tang H (2013) A survey-based exploration of land-system dynamics in an agricultural region of Northeast China. Agricultural Systems 121:106-116. doi:https://doi.org/10.1016/j.agsy.2013.06.006

Yu Q, You L, Wood-Sichra U, Ru Y, Joglekar AKB, Fritz S, Xiong W, Lu M, Wu W, Yang P (2020) A cultivated planet in 2010 – Part 2: The global gridded agricultural-production maps. Earth System Science Data 12:3545-3572. doi:10.5194/essd-12-3545-2020

---

## Author Response (AR2)

Dear Dr. Hanqin Tian and reviewers,

We appreciate for the opportunity to improve our manuscript.

We would like to thank you for the time and effort spent on reviewing. In the resubmitted manuscript, we explicitly address all of the comments. Language was edited thoroughly in preparing the resubmission.

Additionally, as we were concerned about the modest manuscript score on data quality criterion, we improved our data quality after a short communication with the editor (Dr. Hanqin Tian). Specifically, we improved the accessibility and usability of the dataset by providing an alternative data format including selected data summaries, and we extended insights in dataset validity by performing additional comparisons with FAOSTAT structural data which was firstly released two months ago. FAOSTAT structural data includes and significantly extends one of the datasets that we compared in the original manuscript, thus, we reorganized the respective content in the resubmission.

Below you can find our revisions, structured as: **[Comment]** from reviewers, [Response] from authors, and [Change] made in the manuscript (clean mode).

Best regards,

Han Su, on behalf of all co-authors

PhD Candidate

Multidisciplinary Water Management group, University of Twente

**Report #1**

[Comment] The manuscript has addressed the concerns raised during the previous round of review. Editorial suggestions are listed below, but it is up to the author to decide whether to adopt them. This is a nice work, and I acknowledge the authors' efforts to develop the dataset.

[Response] Thank you for your acknowledgment. We appreciate your comments, the time and effort spent on reviewing. All the suggestions are agreed on and implemented.

Besides what was suggested, we were eager to improve our data quality once receiving your review report, due to its only modest rating.

Already in the original manuscript, following the EESD guidelines, we have made the best use of state-of-art publicly available data sources to develop and validate this dataset. The uncertainty issue is explicitly addressed in the current manuscript. Although there obviously is uncertainty, as in any dataset, we do not think current uncertainty will limit the potential applications of our dataset in global studies, particularly since there are no comparable other datasets available. Dataset validity information may guide interpretation in the applications.

Still, we see potentials for data quality improvement regarding data accessibility and comparison with additional dataset. After communicating with the editor, we decided to provide the dataset in an additional data format (netCDF) with selected data summaries. We hope our dataset is more accessible for some potential users with the new data format. We also grasped the opportunity to add an extra comparison with FAOSTAT structural data on farm size which was firstly released just two months ago. The FAOSTAT structural data covers one of the datasets that we compared before, thus, we replaced one of the original comparisons with this new one.

[Change] The description of FAOSTAT structural data and how it covers Lowder et al. (2016), that we compared in the original manuscript, were added in line 255-262. The new comparison with FAOSTAT structural data can be found in line 357-376, Fig. 7 and Fig. A3.

**Technical corrections:**
[Comment] 1. L108-109. Although the two crop maps appear line 118, I would suggest mentioning the name of datasets (GAEZv4 and SPAM2010) in lines 108-109 for clarity. Since many datsets are used in this study, I was confused which datasets the authors mention.

[Response] We agree.

[Change] The names of datasets (GAEZv4 and SPAM2010) were added in line 107.

[Comment] 2. For GAEZv4 and SPAM2010, the indicator is labelled "crop map" in Fig. 1 but it appears to be "harvested area" in Table 1. It would be more readable if a note is added to the figure and table to state these are interchangeable here.

[Response] We agree.

[Change] Notes were added to Fig. 1 and Table 1 showing the "crop map" and "harvested area" are interchangeable.

**[Comment] 3. L435. "SDG2 (Zero Hunger)" is true. But more precisely, it is Target 2.3 of SDG2 ( see https://unstats.un.org/sdgs/metadata/?Text=&Goal=2&Target=2.3 ).**

[Response] We agree.

[Change] The specification "Target 2.3" was added in line 497.

**Report #2**

**[Comment] The authors tried to develop a gridded map of farm-size dataset by harmonizing different sources of data. I agree this is an interesting study and the data developed has potential to improve global/regional simulations. However, the current version is not in good shape.**

[Response] Thank you for your comments which enable us to improve the manuscript. We appreciate the time and effort you spent on reviewing. All the suggestions are agreed on, and implemented.

Besides what was suggested, we were eager to improve our data quality once receiving your review report, due to its only modest rating.

Already in the original manuscript, following the EESD guidelines, we have made the best use of state-of-art publicly available data sources to develop and validate this dataset. The uncertainty issue is explicitly addressed in the current manuscript. Although there obviously is uncertainty, as in any dataset, we do not think current uncertainty will limit the potential applications of our dataset in global studies, particularly since there are no comparable other datasets available. Dataset validity information may guide interpretation in the applications.

Still, we see potentials for data quality improvement regarding data accessibility and comparison with additional dataset. After communicating with the editor, we decided to provide the dataset in an additional data format (netCDF) with selected data summaries. We hope our dataset is more accessible for some potential users with the new data format. We also grasped the opportunity to add an extra comparison with FAOSTAT structural data on farm size which was firstly released just two months ago. The FAOSTAT structural data covers one of the datasets that we compared before, thus, we replaced one of the original comparisons with this new one.

[Change] The description of FAOSTAT structural data and how it covers Lowder et al. (2016), that we compared in the original manuscript, were added in line 255-262. The new comparison with FAOSTAT structural data can be found in line 357-376, Fig. 7 and Fig. A3.

**[Comment] Language editing is strongly recommended. The language should be improved, especially the descriptions in the results section.**

[Response] We agree.

[Change] Language was edited thoroughly in preparing the resubmission, paying special attention to results section.

**[Comment] Another major issue is that a substantial amount of discussion was mixed into the results. These contents should be moved to the discussion section. A typical example is the content in Lines275-280.**

[Response] We agree. Considering this manuscript is a data description paper, we limited the results section to present the data statistics, validation of the datasets, and comparison of the dataset, and moved other content to discussion.

[Change] We moved the explanations on farming systems and farm sizes to discussion section 4.1, line 407-426.

**[Comment] Besides, there are numerous, minor errors, which indicate that the authors didn't pay much attentions on this work.**

[Response] We tried our best to avoid any errors when preparing and revising the manuscript. Unfortunately, there were still some minor errors. We appreciate you pointing them out.

[Change] All the errors found by reviews (as below) and ourselves were fixed.

**Other suggestions:**
**[Comment] 1. Line67, why 2018b is cited before 2018a.**

[Response] Citations and references are managed by EndNote using the official style provided in the ESSD submission guideline. According to the style, the reference is sorted alphabetically not by the citation order. Thus, it is possible to cite 2018b first. But, the Ricciardi et al. (2018a) and Ricciardi et al. (2018b) are special because they are the same study: Ricciardi et al. (2018a) is the data brief paper of Ricciardi et al. (2018b).

We agree citing 2018b before 2018a might be confusing. For this specific case, we cite both Ricciardi et al. (2018a) and Ricciardi et al. (2018b) to avoid potential confusion.

[Change] Ricciardi et al. (2018a) and Ricciardi et al. (2018b) were simultaneously cited throughout the manuscript.

**[Comment] 2. Why a reference at the end of the Line69?**

[Response] The reference was meant as the start of the next sentence.

[Change] We clarified the citation in line 70.

**[Comment] 3. ?Line79-80, the references.**

[Response] We interpret the comment to mean that references were not cited properly, and agree with that. We improved the whole sentence and citations.

[Change] The citations were clarified, lines 84.

**[Comment] 4. Line90, should be "estimate"**

[Response] We agree.

[Change] The sentence was rewritten, line 93-95.

**[Comment] 5. In section 3.2, you actually only talked about the dash line (overall) in Figure 4a. I would suggest to use thin lines for other cases. The figure is distracting and readers' attentions will be directed to the colored lines.**

[Response] We agree. The dark line is firstly described than the colored lines. The figure is not optimal.

[Change] We improved the figure style based on the above suggestion, Fig. 5.

**[Comment] 6. Line268, what observation. Should be revised to be more clear to readers. This is a typical example that the descriptions in this section should be improved.**

[Response] We agree. The "observation" refers to "small farms irrigate a larger share of their area than large farms", which is shown by our dataset and previous studies

[Change] We improved the description in this section for better clarity, line 289-291.

**[Comment] 7. Lines282-283, what do you mean? Not clear.**

[Response] We mean large farms irrigate more when water is scarce.

[Change] We improved the description for better clarity, line 307-309.

**[Comment] 8. For Figure 4, "significant" is not clear to readers, although I know it indicates a water scarcity level higher than moderate. Explanation will be needed in the title. For example, low, moderate… indicate …**

[Response] We agree.

[Change] Explanations on the classification of water scarcity levels were added in the figure title, line 317-319, 418-420.

**[Comment] 9. Suggest to redraw all scatter plots using more professional software (e.g. R). They are not aesthetically attractive.**

[Response] We agree.

[Change] All the scatter plots were redrawn using *seaborn* and *matplotlib.pyplot* package in Python, Fig. 6.

**[Comment] 10. Line312, please cite the source of the data. Which table?**

[Response] We agree.

[Change] The tables were added as supplementary materials ([S6] and [S7]) and cited in the manuscript, line 326.